# Plasma brain-derived tau is an amyloid-associated neurodegeneration biomarker in Alzheimer's disease

Fernando Gonzalez-Ortiz [1,2,33] ✉, Bjørn-Eivind Kirsebom[3,4,5,33], José Contador [6,7,8], Jordan E. Tanley[9], Per Selnes[10], Berglind Gísladóttir[10], Lene Pålhaugen[10], Mathilde Suhr Hemminghyth[11,12,13], Jonas Jarholm[10], Ragnhild Skogseth[14,15], Geir Bråthen [16,17], Gøril Grøndtvedt[16,17], Atle Bjørnerud[18,19,20], Sandra Tecelao[10], Knut Waterloo[3,4], Dag Aarsland[21,22], Aida Fernández-Lebrero[6,7,8,23,24], Greta García-Escobar[7,24], Irene Navalpotro-Gómez[6,7,8,24], Michael Turton[25], Agnes Hesthamar[1], Przemyslaw R. Kac [1], Johanna Nilsson [1], Jose Luchsinger[9], Kathleen M. Hayden [9], Peter Harrison[25], Albert Puig-Pijoan [6,7,24,26], Henrik Zetterberg [1,2,27,28,29,30], Timothy M. Hughes[9], Marc Suárez-Calvet [6,7,8,31], Thomas K. Karikari [1,32,34], Tormod Fladby [5,10,34] & Kaj Blennow [1,2,34]

Staging amyloid-beta (Aβ) pathophysiology according to the intensity of neurodegeneration could identify individuals at risk for cognitive decline in Alzheimer's disease (AD). In blood, phosphorylated tau (p-tau) associates with Aβ pathophysiology but an AD-type neurodegeneration biomarker has been lacking. In this multicenter study (*n* = 1076), we show that brain-derived tau (BD-tau) in blood increases according to concomitant Aβ ("A") and neurodegeneration ("N") abnormalities (determined using cerebrospinal fluid biomarkers); We used blood-based A/N biomarkers to profile the participants in this study; individuals with blood-based p-tau+/BD-tau+ profiles had the fastest cognitive decline and atrophy rates, irrespective of the baseline cognitive status. Furthermore, BD-tau showed no or much weaker correlations with age, renal function, other comorbidities/risk factors and self-identified race/ethnicity, compared with other blood biomarkers. Here we show that blood-based BD-tau is a biomarker for identifying Aβ-positive individuals at risk of short-term cognitive decline and atrophy, with implications for clinical trials and implementation of anti-Aβ therapies.

Alzheimer's disease (AD) is pathologically characterized by amyloid-beta (Aβ) plaques, tau neurofibrillary tangles and brain atrophy[1–3]. While it can take several years for clinical symptoms to manifest in the form of cognitive impairment, the molecular alterations that precede symptom onset can be detected using biomarkers[4–6]. Consequently,

ongoing efforts to enable early disease detection, treatment and sub-grouping of participants in clinical therapeutic trials are including biomarkers in their programs[7] as recommended by the US FDA. In recent years, blood biomarkers that fulfill these requirements have been developed and independently validated across multiple

cohorts[1,2,8]. Plasma Aβ42/Aβ40 ratio and phosphorylated tau (p-tau) biomarkers targeting the epitopes threonine-181 (p-tau181), threonine-217 (p-tau217), and threonine-231 (p-tau231) have each shown high accuracies to identify brain Aβ pathophysiology[2,9–11]. In addition, these biomarkers can monitor longitudinal changes in Aβ pathophysiology among individuals enrolled in clinical trials as well as in those being monitored through various stages of the AD continuum[12,13]. To this end, plasma Aβ42/Aβ40 ratio and p-tau biomarkers are being proposed to prescreen participants for the presence of Aβ pathophysiology prior to inclusion in clinical trials or prescription of anti-Aβ therapies[2,3,7]. If needed, confirmation with Aβ positron emission tomography (PET) or cerebrospinal fluid (CSF) Aβ42/Aβ40 ratio can be performed on those with inconclusive blood biomarker results.

While promising blood biomarkers for brain AD-type Aβ pathophysiology now exist, an unmet need is for a biomarker that tracks neurodegenerative changes specifically in AD[14]. Neurofilament light chain (NfL) in blood is a general marker of neurodegeneration/axonal injury across multiple disorders[15]. Total-tau (t-tau) shows minimal change in AD and has very large overlap between AD, non-AD dementias and control groups[16–18] to be diagnostically useful. A blood biomarker that detects AD-specific neurodegeneration would be crucial to identifying Aβ-positive individuals who are at risk for a more rapid disease course and thus are likely to deteriorate faster in the short term – both biologically and cognitively – than other individuals with similar abnormal Aβ loads. Indeed, a substantial proportion of cognitively normal (CN) Aβ-positive older adults do not develop cognitive decline in years of follow-up[13,16]. Among both Aβ-positive CN and mild cognitive impairment (MCI) individuals, being able to differentiate those at higher risk for future cognitive decline and brain atrophy from others at lower odds is critical especially in the era of anti-Aβ therapies. For instance, it would enable prioritization of those at higher risk for cognitive deterioration and brain atrophy for immediate treatment with the clinically approved anti-Aβ drugs. In line with this, both the 2011 and 2018 updates of the National Institute on Aging-Alzheimer's Association (NIA-AA) research framework proposed criteria to stage the severity of Aβ pathophysiology that focused on the presence or absence of neurodegeneration[19,20]. For example, the 2011 NIA-AA framework proposed a biologically-defined criteria that included the following biomarker categories: (1) Aβ-negative/neurodegeneration-negative (i.e., A-/N-); (2) A+/N-; and (3) A+/N+[19]. The A-/N+ (suspected non-AD pathophysiology [SNAP]) category was excluded because neurodegeneration in this group was thought to emanate from non-AD causes[21]. Although the 2018 NIA-AA update further Supplementary this classification to include tau pathophysiology ("T"), the principle that joint Aβ pathophysiology and neurodegeneration is detrimental in AD remains. In fact, the 2018 framework further hypothesized a pathway that suggested that Aβ pathophysiology drives neurodegeneration to further induce cognitive impairment[19]. CSF and neuroimaging studies evaluating the operationalization of this biomarker staging model have shown that the risk of progression to MCI or AD dementia is highest in the A+/N+ followed by the A+/N- group; the A-/N- and A-/N+ categories recorded similarly little to no risk[21,22].

We recently presented the promising blood-based neurodegeneration biomarker brain-derived tau (BD-tau) that showed higher levels in biomarker-confirmed AD versus non-AD dementia and unaffected controls, including in a cohort with autopsy-verification of diagnosis and in memory clinic cohorts with large heterogeneity of disorders[23]. In the present multicenter study, in four independent cohorts, we examine blood BD-tau associations with longitudinal cognition and AD-signature atrophy rates along the preclinical, MCI and dementia phases of AD. We further evaluate if BD-tau concentrations increase as a function of combined A+ and N+ abnormalities in the AD continuum, and if such an A/N classification can be implemented using blood-based biomarkers to identify those at short-term risk for cognitive decline and atrophy. Furthermore, we assess effects of genetic risk, comorbid conditions and demographic factors including racial self-identity on plasma BD-tau concentrations and clinical performances.

## Results

### Cohort characteristics

We performed cross-sectional and longitudinal analyses in four independent cohorts. *Cohort-1* was the Dementia Disease Initiation (DDI, Norwegian national multicenter study) cohort (n = 364) which longitudinally follows older adults who were CN (n = 221) or had early MCI (n = 143) at study entrance. Cross-sectional analyses in this cohort allowed us to investigate associations of plasma (and CSF) BD-tau with CSF t-tau and Aβ42/β40 ratio, the soluble neurodegeneration and Aβ pathophysiology markers included in the AT(N) framework[2], as well as longitudinal cognition and MRI-derived neurodegeneration evaluations. Of the included participants, n = 255 had repeated cognitive measurements up to 8.25 years (mean = 3.22, SD = 1.56 years). A total of n = 269 participants also had MRI scans; n = 136 had repeated MRIs up to 7.66 years (mean = 2.70, SD = 1.29 years). *Cohort-2* (n = 37; from the Sahlgrenska University Hospital, Gothenburg, Sweden) included CSF A/N (Aβ42, t-tau) biomarker-confirmed AD-dementia participants versus biomarker-negative controls. The research-based cohort results in cohorts 1 and 2 – both cross-sectional and longitudinal – were independently validated in a memory unit clinical cohort (*Cohort-3*; n = 370; from the BIODEGMAR cohort, Hospital del Mar, Barcelona, Spain) with CSF biomarker-determined A/N (Aβ42/β40 ratio, t-tau) categorization. Longitudinal cognitive data from the BIODEGMAR cohort was included (repeated neuropsychological evaluations were performed in n = 209 participants up to 3.35 years (mean=1.57, SD = 0.65). Participant characteristics for cohorts 1, 2 and 3 are summarized in Table 1. Finally, in the United States-based Multi-Ethnic Study of Atherosclerosis (MESA)-MIND pilot study at the Wake Forest University site (*Cohort-4*; Supplementary Table 1, n = 305, 50% self-reported African Americans and 50% non-Hispanic White participants), we evaluated effects of demographic and comorbid conditions on plasma BD-tau levels. A comprehensive description of the cohorts is given in the Methods.

### Aβ-associated concentrations of plasma BD-tau across the clinical AD continuum

We previously demonstrated that plasma BD-tau levels are higher in autopsy-verified AD versus non-AD dementias, and associate with Aβ plaque counts[23]. Here, we hypothesized that plasma BD-tau will increase according to in vivo Aβ pathophysiology if it is an AD-associated neurodegeneration marker. To investigate this, we grouped participants according to joint A and/or N abnormalities (A+/N- or A+/N+), as compared with participants with normal biomarkers (e.g., A-/N-). Here, A-/N- were CN in cohort 1, CN in cohort 2 and mixed CN, MCI, and Dementia in cohort 3. We used CSF Aβ42/40 ratio (cohort-1 and 3), Aβ42 (cohort-2), and t-tau (cohorts 1–3) as the standard A and N biomarkers because, similar to their plasma alternatives, they reflect soluble biomarker changes that tend to become abnormal earlier in AD than neuroimaging biomarkers[24].

In cohort 1, plasma BD-tau concentrations were significantly higher in the A+/N+ group (p < 0.001) compared with the A-/N- CN cases, but not the A+/N- group despite stepwise increases in the z-scores from A-/N- to A+N- and then A+/N+ (Fig. 1). The results remained when the participants were divided into separate diagnostic groups i.e., CN and MCI (Supplementary Table 2 and Supplementary Fig. 1). While plasma t-tau and NfL levels were significantly higher in all A+/N+ versus A-/N- groups, the largest difference in biomarker concentrations were demonstrated for plasma BD-tau (mean difference [mdiff] = +0.98 SD, Supplementary Table 2, and Supplementary Fig. 1).

These findings were corroborated in the dementia stage (cohort-2) where serum BD-tau, but not t-tau, concentrations were

**Table 1 | Between-group comparisons of selected demographic variables in each cohort**

| Cohort 1 (DDI cohort) | | | | | | | |
|---|---|---|---|---|---|---|---|
| | **A/T/N groups (n)** | | | **Statistical tests** | | | |
| | **CN A-/ N- (157)** | **Cases A +/N- (45)** | **Cases A+/N + (162)** | **F/$\chi^2$/$\eta^2$/$\eta p^2$/(p)** | **A-/N- vs A+/N-** | **A-/N- vs A+/N+** | **A+/N- vs A+/N+** |
| Age Mean (SD) | 59.4 (9.0) | 66.4 (7.6) | 68.2 (7.6) | F = 47.1, $\eta^2$ = 0.21, (**<0.001**) | c**<0.001** | c**<0.001** | cn.s. |
| Female n (%) | 87 (55.4) | 33 (73.3) | 79 (48.8) | $\chi^2$ = 8.6, (**<0.05**) | a | a | a |
| Recruited as Controls n (%) | 52 (33.1) | 4 (8.9) | 13 (8.0) | b | a | a | a |
| SCD n % | 105 (66.9) | 19 (42.2) | 33 (20.4) | b | a | a | a |
| MCI n % | 0 () | 22 (48.9) | 116 (71.6) | b | a | a | a |
| CERAD Recall Mean (SD) | 7.5 (1.7) | 5.3 (2.7) | 3.8 (2.8) | F = 99.9.1, $\eta p^2$ = 0.36, (**<0.001**) | d**<0.001** | d**<0.001** | d**<0.01** |
| TMT-B seconds Mean (SD) | 75.3 (23.1) n=156 | 119.3 (81.4) n=41 | 140.0 (84.9) n=155 | F = 66.6.1, $\eta p^2$ = 0.28, (**<0.001**) | d**<0.001** | d**<0.001** | dn.s. |
| APOE-ε4+ n (%) | 45 (29.4) n=153 | 27 (62.8) n=43 | 115 (73.2) n=157 | $\chi^2$ = 61.7, (**<0.001**) | a | a | a |
| **Cohort 3 (Biodegmar cohort)** | | | | | | | |
| | **A-/N- (111)** | **A+/N- (79)** | **A+/N+ (180)** | **F/$\chi^2$/$\eta^2$/(p)** | **A-/N- vs A+/N-** | **A-/N- vs A+/N+** | **A+/N- vs A+/N+** |
| Age Mean (SD) | 70.8 (6.8) | 74.9 (4.4) | 73.9 (5.3) | F = 14.7, $\eta^2$ = 0.07, (**<0.001**) | c**<0.001** | c**<0.001** | cn.s. |
| Female n (%) | 49 (44.1) | 45 (57.0) | 117 (65.0) | $\chi^2$ = 12.2, (**<0.05**) | a | a | a |
| SCD n % | 17 (15.3) | 6 (7.6) | 3 (1.7) | b | a | a | a |
| MCI n % | 53 (47.7) | 27 (34.2) | 57 (31.7) | b | a | a | a |
| Dementia n % | 41 (36.9) | 46 (58.2) | 120 (66.7) | b | a | a | a |
| MMSE Mean (SD) | 24.0 (5.4) | 20.2 (6.1) | 20.4 (4.5) | F = 18.7, $\eta^2$ = 0.09, (**<0.001**) | c**<0.001** | c**<0.001** | cn.s. |
| CDR Median [IQR] | 0.5 [0.5;1.0] | 1.0 [0.5;2.0] | 1.0 [0.5;2.0] | F = 5.0, $\eta^2$ = 0.03, (**<0.01**) | cn.s. | c**<0.01** | cn.s. |
| APOE-ε4+ n (%) | 14 (16.7) | 32 (51.6) | 89 (61.0) | $\chi^2$ = 43.0, (**<0.001**) | a | a | a |
| **Cohort-2 (Gothenburg cohort)** | | | | | | | |
| | **CN A-/N- (9)** | **Dementia A+/N+ (28)** | | **t / ft / d(p)** | e | e | e |
| Age Mean (SD) | 56.9 (17.5) | 73.2 (8.3) | | t = -2.7, d = 1.47, (**<0.05**) | e | e | e |
| Female n (%) | 5 (55.56) | 16 (57.14) | | ft = 1.06, (n.s.) | e | e | e |

Demographic characteristics and between group differences in cohorts 1 to 3. Significant p-values (p < 0.05) highlighted in bold. All statistical tests were two-sided.

*A+/-* positive or negative CSF marker for amyloid plaques, *N+/-* positive or negative marker for neurodegeneration, *CN* Cognitively Normal, *SCD* Subjective Cognitive Decline, *MCI* Mild Cognitive Impairment, *CERAD recall* Consortium to Establish a Registry for Alzheimer's Disease (CERAD) verbal memory recall, *TMT-B* Trail-Making Test B; *MMSE* Mini Mental Status Examination, *CDR* Clinical Dementia Rating, *APOE-ε4+* Apolipoprotein ε 4 allele carriers, *SD* standard deviation, *n* number of cases, % percentage, *F* F statistic, *t*, welch two-sample t-test, *$\chi^2$*, chi square statistic, *$\eta^2$*, eta-squared, *$\eta p^2$*, partial eta-squared *d* cohen´s d; *ft* Fischer´s Exact Test statistic; *vs* versus.

aNo post-hoc comparisons performed.
bNo value.
cANOVA post-hoc comparisons (unadjusted).
dANCOVA post-hoc comparisons (unadjusted).
eNo data/comparisons performed.

significantly higher in A+/N+ individuals with dementia as compared with the A-/N- CN (p < 0.001, Fig. 1 and Supplementary Table 2). Here, the between-group difference for CN versus AD-dementia was substantially larger (mdiff = +3.73 SD) than what was demonstrated for the pre-dementia participants in cohort 1 (mdiff = +0.98 SD). Contrarily, serum t-tau showed a more comparable difference between controls and cases in cohort-2 (mdiff = +0.67 SD) as in cohort-1 (mdiff = +0.75 SD). Note that in cohort-2 (and cohort-3 below), BD-tau was measured in serum and has equivalent diagnostic performances in serum versus ethylenediaminetetraacetic acid (EDTA) plasma, despite the absolute levels being lower in serum[25] as also shown for t-tau and NfL[26].

## Aβ association of plasma BD-tau with CSF t-tau in the AD continuum

In cohort-1, a strong correlation between CSF t-tau and CSF BD-tau (r = 0.91, p < 0.001; Supplementary Fig. 2A) across A/N groups was found, suggesting that these markers are highly equivalent in CSF. The correlation between CSF t-tau and CSF NfL was comparatively weaker (r = 0.58, p < 0.001; Supplementary Fig. 2B), indicating that they largely reflect different aspects of neurodegeneration in AD. The correlation of plasma BD-tau with CSF t-tau in cohort-1 was more modest (r = 0.42, p < 0.001; Supplementary Fig. 3A), but comparatively stronger than for plasma t-tau with CSF t-tau (r = 0.23, p < 0.001; Supplementary Fig. 3B) and plasma NfL with CSF t-tau (r = 0.33, p < 0.001; Supplementary

Fig. 3C). Importantly, the strength of the correlation between plasma BD-tau with CSF t-tau tended to increase according to concomitant A/N positivity but only the A+/N+ group showed significance; correlation of plasma t-tau and NfL with CSF t-tau did not follow the A/N classification (Supplementary Fig. 1 and 4). The strength of correlation of plasma BD-tau with CSF t-tau was similar to what has been shown for p-tau181, p-tau217, and p-tau231[9,10,27].

In cohort-2 which included only A-/N- and A+/N+ participants, serum BD-tau was strongly associated with CSF t-tau ($r = 0.91$, $p < 0.001$; Supplementary Fig. 3D) but serum t-tau and CSF t-tau were not correlated ($r = 0.02$, $p = 0.903$; Supplementary Fig. 3E). Thus, these findings suggest that in the AD continuum CSF and blood BD-tau are correlated in a manner that is Aβ-associated. Moreover, these results correspond with our previous finding that plasma t-tau and BD-tau do not correlate well[23].

## Association of plasma BD-tau with AD-signature atrophy rate and cognitive decline

In cohort-1, plasma BD-tau (Fig. 2A), but not plasma t-tau (Fig. 2B) or NfL (Fig. 2C), was significantly associated with future AD meta-Region of Interest (ROI) atrophy ($b = -0.06$, $p < 0.01$; Supplementary Table 3). Moreover, in cohort−1, plasma BD-tau was associated with baseline performance on both the Consortium to Establish a Registry for Alzheimer's Disease (CERAD) verbal memory recall and the Trail Making Test-B (TMT-B, Fig. 3A); $b = -0.16$, $p < 0.001$ and $b = 0.12$, $p < 0.01$ respectively, as well as the longitudinal worsening in performance on both tests over up to 8 years later ($b = -0.05$, $p < 0.05$; $b = 0.04$, $p < 0.001$ respectively, Supplementary Table 3). Plasma t-tau showed weaker but significant associations with baseline ($b = -0.10$, $p < 0.05$) and longitudinal ($b = -0.03$, $p < 0.01$) CERAD verbal memory recall, but the confidence intervals were wider (Supplementary Table 3). However, no significant associations between plasma t-tau and TMT-B performance were observed (Fig. 3B and Supplementary Table 3). Plasma NfL was not associated with baseline cognitive performance on either CERAD verbal memory recall or TMT-B (Fig. 3C), but it was associated with future decline in both tests ($b = -0.03$, $p < 0.05$; $b = 0.03$, $p < 0.05$ respectively, Supplementary Table 3). Together, only BD-tau was associated with both baseline and cross-sectional performances on the CERAD and TMT-B tests.

## Validation in a memory clinic cohort

In cohort-3, the strong associations of CSF BD-tau and t-tau in cohorts 1 and 2 were replicated ($r = 0.92$, $p < 0.001$; Supplementary Fig. 2C), so was the correlation of serum BD-tau with CSF t-tau ($r = 0.38$, $p < 0.001$; Supplementary Fig. 3F). Moreover, similar to cohort−1, a weaker correlation was found between plasma t-tau and CSF t-tau ($r = 0.25$, $p < 0.001$; Supplementary Fig. 3G). However, no significant correlation between plasma NfL and CSF t-tau was observed ($r = 0.12$, $p = 0.126$; Supplementary Fig. 3H). In cohort 3, all A/N groups comprised different clinical stages ranging from individuals experiencing subjective cognitive decline (SCD) to MCI and dementia cases. Here, both the A

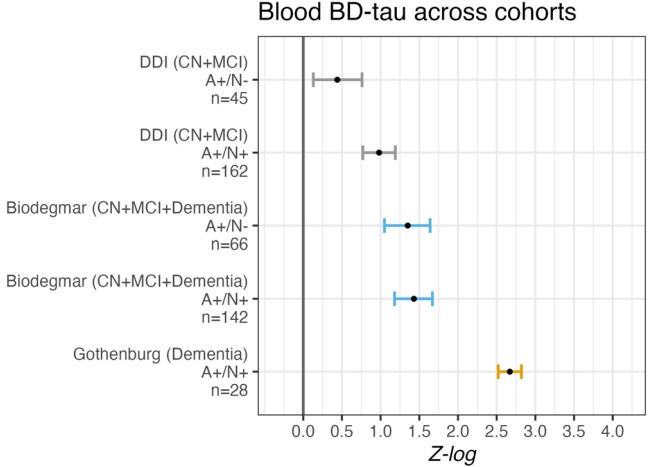

**Fig. 1 | Blood-based brain-derived tau (BD-tau) mean differences across cohorts 1 to 3, compared to A-/N- cases.** This forest plot displays the mean elevation of blood-based BD-tau according to cerebrospinal fluid (CSF) determined amyloid (A) and neurodegeneration (N) status in the Alzheimer's Disease (AD) continuum across cohorts 1 through 3. Point estimates are log-transformed and standardized (Z-log), expressed as standard deviations relative to A-/N- cases (represented by a gray vertical bar, normalized to mean 0 as the reference). Error bars denote 95% confidence intervals. All statistical tests were two-sided and unadjusted for multiple comparisons. For cohort 1 (Dementia Disease Initiation (DDI)), A-/N- were cognitively normal (CN, $n = 157$). For cohort 2 (University of Gothenburg), A-/N- were CN (A-/N-, $n = 9$). For cohort 3 (Biodegmar, Hospital del Mar, Barcelona memory clinic cohort), A-/N- ($n = 111$) were mixed CN with Subjective Cognitive Decline, Mild Cognitive Impairment (MCI) and dementia.

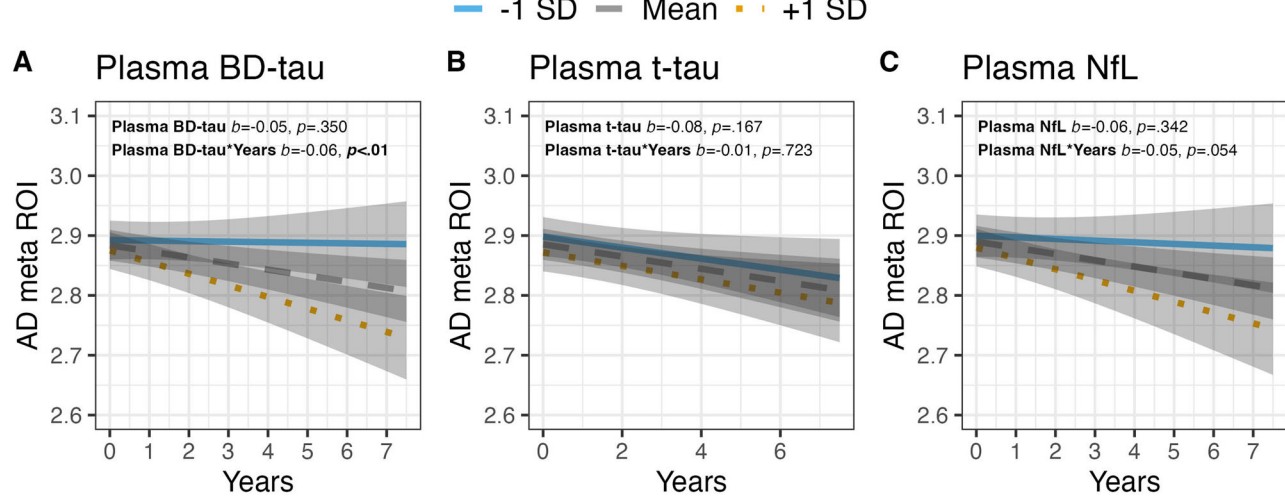

**Fig. 2 | Associations between baseline blood-based BD-tau, t-tau and NfL with baseline and longitudinal MRI-derived Alzheimer's disease meta region of interest (AD meta ROI). A–C** show associations of blood-based neurodegeneration markers with baseline and longitudinal AD meta ROI. All statistical tests were two-sided, and unadjusted for multiple comparisons. All plots display model predictions generated with the "ggeffects" R package. The lines display associations between the biomarker at −1SD, Mean and +1SD and the dependent variable at baseline and over time.

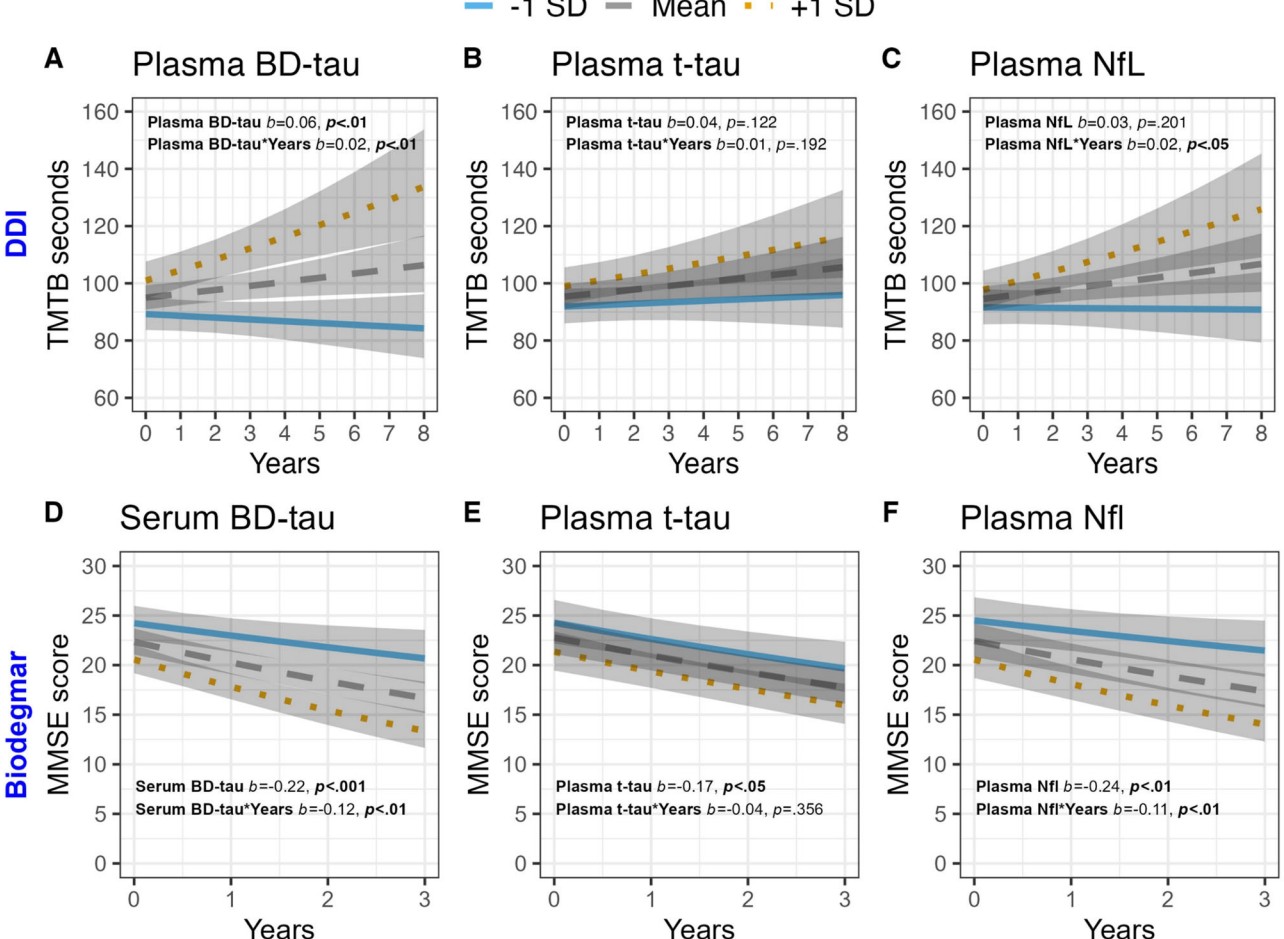

**Fig. 3 | Associations between baseline blood-based BD-tau, t-tau and NfL with baseline and longitudinal cognitive performance.** A–C show associations of neurodegeneration markers in blood with Trail-Making Test B (TMT-B) performance in cohort 1 (Dementia Disease Initiation (DDI)). D–F show associations of neurodegeneration markers in blood with Mini Mental Status Examination (MMSE) scores in cohort 3 (Biodegmar, Hospital del Mar, Barcelona memory clinic cohort). All statistical tests were two-sided, and unadjusted for multiple comparisons. All plots display model predictions generated with the "ggeffects" R package. The lines display associations between the biomarker at −1SD, Mean and +1 SD and the dependent variable at baseline and over time.

+/N- (mdiff = +1.35 SD, $p < 0.001$) and A+/N+ (mdiff = +1.43 SD, $p < 0.001$) groups had higher concentrations of serum BD-tau as compared to A-/N-. Thus, this in part replicates findings from cohorts 1 and 2, but where the statistical significance of the higher BD-tau levels was not limited to the A+/N+ group. Moreover, the relative difference between A+ groups and A- groups were greater in this cohort composed mainly by individuals at MCI to dementia stages of cognitive decline as compared to cohort−1 (which included primarily CN and early MCI individuals) but smaller than in cohort-2 which consisted of exclusively dementia participants versus CN (Fig. 1 and Supplementary Table 2). Considering these results, we carried out additional post-hoc analyses on the A+/N+ and A+/N- groups in cohort 1 & 3 for blood BD-tau. In cohort 3, serum BD-tau concentrations were similar in the A+/N+ ($p = 0.783$) ($M = 15$, $SD = 13.3$) and A+/N- ($M = 14.6$ $SD = 7.9$) group. In contrast the A+/N+ ($M = 12.1$, $SD = 14.8$) group in cohort 1 had significantly higher plasma BD-tau concentrations as compared to the A+/N- group ($M = 6.9$, $SD = 5.8$, $p < 0.001$). Furthermore, whereas cognitive performance was similar in the A+/N- and A+/N+ groups in cohort 3, the A+/N+ group in cohort 1 had worse cognitive performance than the A+/N- groups (Table 1). The A-/N- group in cohort 3 (Biodegmar) comprised both cognitively unimpaired (CN, SCD) and impaired (CI, MCI and dementia) cases. But we found no significant differences in serum BD-tau concentrations between these CN ($n = 15$) and CI ($n = 78$) cases ($t = 0.44$, $p = 0.663$) who were A-/N-. As for

differences between CN and CN in Cohort-3 Aβ+ group, only $n = 4$ participants were CN and thus no reliable statistical analyses could be performed.

One possible explanation for this is that in early stages of the AD continuum, namely preclinical and prodromal AD (cohort 1), Aβ and neurodegeneration are required to increase levels of BD-tau in blood, while in later stages, MCI and dementia (cohort 3), the presence of Aβ pathology can increase blood BD-tau in absence of N positivity in CSF. Secondarily, factors such as blood-brain barrier breakdown might nevertheless lead to elevated blood levels of BD-tau in CSF T-tau negative cognitively impaired individuals. To further assess blood BD-tau in non-AD cognitive impairment, we sourced additional non-AD (Aβ negative MCI) cases ($n = 143$) from cohort 1 along with cognitively impaired non-AD cases (MCI/Dementia) from cohort 3 ($n = 35$). The comparison between the AD and non-AD groups in cohort 1 and 3 (Supplementary Fig. 4) shows significantly higher levels of blood-based BD-tau in the AD group ($p < 0.001$ in both cohorts) in agreement to previously reported results and similar to CSF T-tau[23]. The findings in both cohorts support the role of plasma/serum BD-tau as an AD-associated neurodegeneration marker. Our findings suggest that clinical progression enhances blood BD-tau association to Aβ pathology and neurodegeneration-dependent increases of blood BD-tau may be additionally modulated to the degree or severity of cognitive impairment.

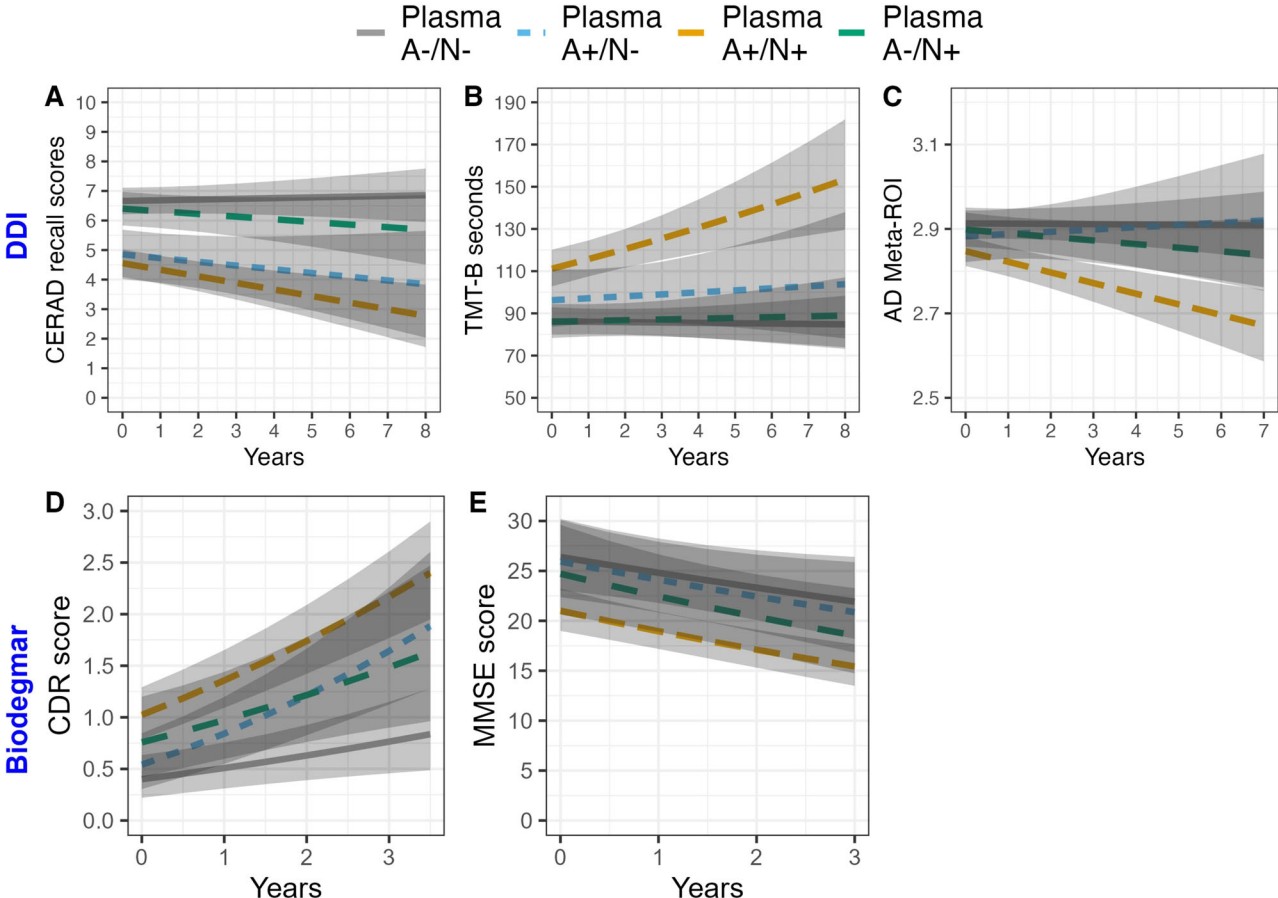

**Fig. 4 | Associations between blood-based A/N groups with baseline and longitudinal cognitive performance and MRI-derived Alzheimer's disease meta region of interest (AD meta ROI). A**, **B** Show the associations between A/N groups based on p-tau181 (A) and BD-tau (N) in blood with baseline and longitudinal cognitive performance (CERAD verbal memory recall and Trail Making Test-B (TMT-B)) in cohort 1 (Dementia Disease Initiation (DDI)). As compared to A-/N-, both A+/N- and A+/N+ profiles showed worse performance at baseline for CERAD, but only A+/N+ had worse performance for TMT-B. However, only A+/N+ profiles were associated with worsening over time for both tests. **C** Shows A/N groups associations with baseline and longitudinal AD meta-ROI, here only A+/N+ showed lower AD meta-ROI at baseline and increased atrophy over time. **D**, **E** Show the associations between A/N groups with baseline and longitudinal cognitive performance (Clinical Dementia Rating (CDR) and Mini Mental Status Examination (MMSE)) in cohort 3 (Biodegmar, Hospital del Mar, Barcelona memory clinic cohort). Here only A+/N+ profiles were associated with poorer scores on MMSE and CDR at baseline. While both A+/N- and A+/N+ showed higher CDRs over time, no longitudinal associations were seen for MMSE in any group. All plots display model predictions generated with the "ggeffects" R package. The lines display associations between the biomarker at -1SD, Mean and +1 SD and the dependent variable at baseline and over time. For more details see Supplementary Table 5.

In cohort-3, higher serum BD-tau was significantly associated with worse baseline MMSE scores ($b = -0.22$, $p < 0.001$, Fig. 3D) and higher Clinical Dementia Ratings (CDRs) ($b = 0.10$, $p < 0.001$), as well as future worsening in both measures ($b = -0.12$, $p < 0.01$ and $b = 0.04$, $p < 0.01$ respectively). Similar associations between plasma NfL and baseline MMSE ($b = -0.24$, $p < 0.01$, Fig. 3F) and future decline ($b = -0.11$, $p < 0.01$) were observed. However, associations between plasma NfL and CDRs were only significant at baseline ($b = 0.15$, $p < 0.001$), but not over time ($b = 0.02$, $p = 0.190$). For plasma t-tau, weaker, albeit significant associations were observed with worse baseline MMSE ($b = -0.17$, $p < 0.05$, Fig. 3E) and CDRs ($b = 0.08$, $p < 0.05$), no significant associations with future worsening in these measures were observed ($b = -0.04$, $p = 0.356$ and $b = 0.02$, $p = 0.068$); Supplementary Table 3.

Combining plasma p-tau and BD-tau operationalizes the A/N classification in blood We evaluated the use of blood p-tau and BD-tau as A and N markers respectively. We used the Youden index approach to generate within-cohort cut-offs for blood BD-tau and p-tau181 positivity in cohorts 1 and 3 (Supplementary Table 4) using the CSF determined CSF A-/N- and A+/N+ as the standard of truth. In cohort-1, this produced the expected A-/N- ($n = 128$), A+/N- ($n = 36$) and A+/N+

($n = 120$) groups, but additionally also a A-/N+ ($n = 77$) group. Longitudinal analyses on cognitive measures in cohort−1 showed that both the A+/N- ($b = -0.59$, $p < 0.001$) and A+/N+ ($b = -0.69$, $p < 0.001$) groups had worse memory at baseline compared with the A-/N- group (Fig. 4A). However, only the A+/N+ showed significant worsening over time ($b = -0.08$, $p < 0.01$) compared with the A-/N- group. Interestingly, the A-/N+ group did not differ from the A-/N- at baseline nor over time. For TMT-B (Fig. 4B), only the A+/N+ group had significantly worse performance at baseline ($b = 0.50$, $p < 0.001$) and longitudinally compared with the A-/N- participants ($b = 0.08$, $p < 0.01$). Moreover, only the A+/N+ had significantly lower AD meta-ROI thicknesses (Fig. 4C, $b = -0.41$, $p < 0.01$) at baseline, and showed atrophy over time ($b = -0.15$, $p < 0.01$) as compared with the A-/N- group (Supplementary Table 5).

In cohort 3, the blood-based markers produced the same groups as in cohort 1. Namely, A-/N- ($n = 30$), A+/N- ($n = 22$), A+/N+ ($n = 68$) and A-/N+ ($n = 14$). Here, baseline MMSE scores (Fig. 4E) were similar to the A-/N- group at baseline for both the A+/N- ($b = -0.05$, $p = 0.851$) and the between the A-/N+ group ($b = -0.19$, $p = 0.538$) but significantly worse in the A+/N+ as compared to the A-/N- cases ($b = -0.65$, $p < 0.01$). Here, no difference between groups on MMSE decline over time was

observed. For CDRs (Fig. 4D), only the A+/N+ cases had higher CDRs at baseline ($b = 0.38$, $p < 0.001$) compared with the A-/N- group. However, both the A+/N- ($b = 0.10$, $p < 0.05$) and A+/N+ ($b = 0.07$, $p < 0.05$), but not the A-/N+ group ($b = 0.04$, $p = 0.474$), had increases in CDRs over time compared to A-/N- (Supplementary Table 5).

### Associations with demographic, genetic, and comorbid risk factors

In cohort−1, age was more strongly associated with plasma NfL ($r = 0.514$, $p < 0.001$) than plasma BD-tau ($\rho = 0.196$, $p < 0.001$) but did not correlate with plasma t-tau ($\rho = 0.048$, $p = 0.362$). A subset of cases had estimated glomerular filtration rate (eGFR) measures available ($n = 281/366$). Reduced eGFR, meaning worse kidney function, was also more strongly associated with plasma NfL ($\rho = -0.257$, $p < 0.001$) than plasma BD-tau ($\rho = -0.180$, $p < 0.01$), but not with plasma t-tau ($\rho = 0.029$, $p = 0.613$). However, eGFR and age are known to correlate[28]. Therefore, using multiple linear regression (with log transformed variables due to non-normality) we assessed if the age association may be partially or fully explained by eGFR, or vice versa. These analyses showed that, when controlling for the effects of eGFR on plasma BD-tau ($b = -0.131$, $p < 0.05$), no significant effects of age remained ($b = 0.076$, $p = 0.243$). On the other hand, for plasma NfL, significant effects of both eGFR ($b = -0.175$, $p < 0.01$) and, more prominently, age ($b = 0.446$, $p < 0.001$) were demonstrated. While higher plasma BD-tau was observed for $APOE$ ε4 carriers (+0.31 SD compared with non-carriers, $p < 0.01$), this effect disappeared when accounting for the effect of A/N group status, suggesting that A/N biomarker status modulates BD-tau levels more strongly than $APOE$ ε4 carrier-ship. Similar results were also found for plasma t-tau (+0.34 SD as compared with non-carriers, $p < 0.01$) which also disappears after controlling for A/N group status. For plasma NfL, no univariate association with $APOE$ ε4 carrier status was observed (+0.08 SD, $p = 0.427$).

In cohort-3, age was also more strongly associated with plasma NfL ($r = 0.377$, $p < 0.001$) than serum BD-tau ($\rho = 0.151$, $p < 0.01$), and plasma t-tau ($\rho = 0.194$, $p < 0.01$). On the other hand, $APOE$ ε4 carriers showed higher plasma t-tau ($b = 0.340$, $p < 0.05$) levels, although this effect also disappeared after controlling for A/N group status. Neither serum BD-tau ($b = 0.141$, $p = 0.293$) or NfL ($b = -0.156$, $p = 0.346$) showed univariate associations with $APOE$ ε4 carrier status. No sex differences were found in serum BD-tau ($b = 0.108$, $p = 0.356$), plasma t-tau ($b = 0.039$, $p = 0.797$) or plasma NfL ($b = 0.116$, $p = 0.468$) levels.

In the biracial cohort 4, regression models (logistic bi/multinomial or linear depending on the response variable) with plasma BD-tau or NfL as the predictor were fitted to assess associations with pertinent response variables. When applicable, years of education, age, gender, and race were included as covariates. Here, neither age ($b = -.002$, $p = 0.691$), eGFR ($b = -.013$, $p = 0.537$), $APOE$ ε4 carrier status ($b = 1.00$, $p = 0.268$) nor race (non-Hispanic White versus African American; $b = 1.00$, $p = 0.498$) was associated with plasma BD-tau. In contrast, higher age ($b = 0.067$, $p < 0.001$) and reduced eGFR ($b = -0.166$, $p < 0.001$) were associated with higher plasma NfL concentrations, but not with $APOE$ ε4 carrier status ($b = 0.994$, $p = 0.636$) nor self-identified race ($b = 0.999$, $p = 0.676$). We found no associations between plasma BD-tau and pertinent comorbidities or risk factors (Low Density Lipoprotein (LDL), High Density Lipoprotein (HDL), total cholesterol, systolic or diastolic blood pressure, body mass index/weight or diabetes mellitus). However, for plasma NfL, a positive association was observed with higher systolic blood pressure ($b = 0.075$, $p < 0.05$). Compared with non-diabetics, a positive and significant association was found between higher NfL concentrations and cases with treated diabetes ($b = 0.979$, $p < 0.05$) but not, for cases with impaired fasting glucose levels ($b = 0.950$, $p = 0.220$) or untreated diabetes ($b = 0.998$, $p = 0.624$).

## Discussion

In this study, we have shown that plasma/serum BD-tau is an AD-associated marker which levels are elevated in the presence of Aβ pathology, predicts cognitive decline and associate with AD-type meta-ROI MRI signatures in the AD continuum. In the preclinical and early MCI phases (cohort 1), blood BD-tau followed CSF t-tau elevation, whereas in later MCI and dementia phases, blood BD-tau elevation was apparent regardless of CSF t-tau levels (cohort 3). Here, blood-brain barrier breakdown in more advanced cases of AD may contribute to elevation of blood BD-tau seen in A+/N- cases in cohort 3. Indeed, evidence of different AD subtypes have recently been reported, where one suggested subtype was marked by evidence of blood-brain barrier dysfunction, lower CSF t-tau and nevertheless high risk of clinical progression[29,30]. Moreover, we showed that BD-tau was not elevated in cognitively impaired non-AD cases and suggest that blood-BD-tau is prominently associated with clinical severity in the AD continuum.

Using blood A/N definitions we showed that the A+/N+ participants in both cohorts had the fastest longitudinal cognitive decline and atrophy profiles, followed by the A+/N- individuals while the double negatives (A-/N-) and SNAP (A-/N+) had the lowest odds for future cognitive decline and neurodegeneration. Together, these results show that plasma BD-tau associates with brain neurodegenerative signatures that interact with Aβ pathophysiology to synergistically drive cognitive decline. We further demonstrate that combining plasma p-tau and BD-tau as blood biomarkers of A and N respectively provides an approach to stage the severity of Aβ pathophysiology and the risk for near-term cognitive decline across the AD continuum. These findings allow operationalization in blood of the two-feature A/N biomarker classification originally proposed in the 2011 NIA-AA research framework[19]. These results will be critical in the era of disease modifying therapies; towards distinguishing Aβ+ participants into different priority groups to be prescribed anti-Aβ therapies according to their likelihood of future cognitive decline and brain atrophy.

Neuroimaging and CSF biomarker studies have demonstrated that Aβ pathophysiology and neurodegeneration work in tandem to drive cognitive decline in AD. For instance, Jack et al.[22], showed that longitudinal increases in Aβ pathophysiology were limited to those with abnormal versus normal Aβ-PET scans at baseline, and the annualized accumulation rate was greater in the A+/N+ and A+/N- groups compared with the Aβ-negative participants (i.e., A-/N- and A-/N+ groups). These findings show that the presence of AD-type neurodegeneration potentiates temporal changes in Aβ pathophysiology and allows to classify Aβ-positive individuals according to who is most likely to decline cognitively in a short time frame. Furthermore, the longitudinal rate of atrophy was greatest in the A+/N+ versus the A+/N- and A-/N- participants, suggesting that Aβ pathophysiology drives neurodegeneration in AD. Other studies using CSF biomarkers have shown that Aβ pathophysiology and neurodegeneration act synergistically to drive cognitive decline; individuals with high CSF t-tau and low Aβ42 or Aβ42/40 ratio show more rapid cognitive decline than those with normal CSF t-tau and normal Aβ profiles[31,32]. In the current study, we showed that these findings can be extended to blood by using BD-tau. Plasma/serum BD-tau increased according to combined A and N abnormalities, indicating that this biomarker reflects neurodegeneration associated with brain amyloidosis. Since plasma p-tau variants have been shown in multiple antemortem and postmortem studies to associate with Aβ pathophysiology across the AD continuum[9,10,12,27,33] and demonstrate higher robustness than plasma Aβ42/40 ratio[2,24], we sought to establish a blood biomarker-based two-feature A/N classification system by combining it with BD-tau. In agreement with the CSF studies that showed that concomitant abnormalities in Aβ42/40 ratio and t-tau were detrimental to cognitive decline and brain atrophy, the plasma p-tau +/BD-tau+ and the p-tau+/BD-tau- cases had much steeper longitudinal cognitive decline and atrophy profiles. These results indicate that combining BD-tau and plasma p-tau is a way to group Aβ-positive

individuals according to their likelihood of cognitive decline. Notably, cross-sectional cognitive performance was worst in the p-tau+/BD-tau+ followed by the p-tau+/BD-tau- cases, meaning that those with abnormal levels of both plasma p-tau and BD-tau are at much higher odds of cognitive decline in the short term. Importantly, our results show that neurodegeneration positivity without abnormal brain Aβ (plasma p-tau-/BD-tau+) is not related to worsening cognition or atrophy, as demonstrated in two cohorts.

Previous studies have reported that CSF t-tau is associated with Aβ pathophysiology but CSF NfL is not[34,35]. Our results extend these observations to blood; blood BD-tau showed larger effect sizes to separate the A/N groups, meaning that it is more responsive to the pathophysiological presence of Aβ and neurodegeneration than both NfL and t-tau in blood. Moreover, the modest correlation between CSF t-tau and NfL supports the previous observations that CSF NfL is not an AD-specific neurodegeneration biomarker[36]. To the contrary, CSF BD-tau and CSF t-tau had very strong correlations of >0.9 in both cohorts 1 and 3, indicating that the biomarkers are largely interchangeable. The strength of correlation was similar across the A/N groups, meaning it is related to the dual presence of Aβ pathophysiology and neurodegeneration.

In cohorts 1–3, blood BD-tau levels increased according to A/N positivity. However, the z-scores also followed the severity of cognitive impairment, with the largest effect size in the dementia cohort (cohort-2), followed by the MCI and dementia cohort (cohort-3) and lowest in the CN and early cognitive decline cohort (cohort-1). These results suggest that the interplay of A/N biomarkers in CSF and for that matter blood BD-tau concentrations is reflected in cognitive status.

Research frameworks and diagnostic guidelines place SNAP individuals as outside the AD continuum[19,20]. In our analysis, the cognitive trajectories of the A-/N+ participants did not differ from the A-/N- group similar to what it has been reported for SNAP profiles in CSF[37].

In this study, we used CSF t-tau and Aβ42/40 ratio to define N and A abnormalities respectively, informed by the understanding that blood and CSF reflect the similar pools of soluble tau variants[38]. Although CSF Aβ42/40 ratio and Aβ PET strongly correlate[39], the N biomarkers – t-tau and NfL in CSF as well as MRI signatures of brain atrophy and PET evaluation of glucose metabolism – have poorer concordances[34,35,40]. To this end, operationalization of the AT(N) criteria has encountered challenges from the non-interchangeability of CSF and imaging biomarkers[41].

The lack of, or weak, associations of blood BD-tau with eGFR and other conditions supports the concept that the BD-tau assay, which uses an antibody binding to a contiguous amino acid sequence at the exon 4–5 junction of the *MAPT* gene, avoids blood-based tau from peripheral sources including the kidney. Indeed, the predominant form of tau in peripheral tissues tends to be of the big tau isoform[42,43]. The blood NfL assay which was not designed to separate out brain and peripheral sources of this protein showed a rather stronger association with eGFR across cohorts. Furthermore, the lack of a strong association of blood BD-tau with age and *APOE* ε4 suggests that the biomarker levels do not substantially increase with biological aging or *APOE* ε4 carriership, meaning age and *APOE* genotype adjustment will not be required in the clinical application of this biomarker including the generation of cut-off values. These results contrast those for blood NfL which requires age stratification in the development of cutoff values[44].

Our findings are directly relevant in the era of anti-Aβ disease-modifying therapies as they allow Aβ+ individuals who may not develop (further) cognitive decline during their lifetime to be distinguished from those at high likelihood of future deterioration. Currently, confirmation of abnormal Aβ pathophysiology is required before patients can be prescribed these treatments. However, as shown in this study, Aβ-positive individuals can be separated into those with high, intermediate, and low risk for future cognitive

deterioration based on their neurodegeneration profile. Implementation of this approach, operationalizable by using highly accessible blood biomarkers, would enable Aβ-positive older adults who are at the highest risk for cognitive decline in the short term to be prioritized for immunotherapy whilst sparing or further monitoring those with the least risk for cognitive decline. Since substantial proportions of pre-dementia Aβ-positive older adults do not show cognitive worsening at follow-up[13,31,32], there is clinical benefit in being able to identify those with the highest likelihood for further cognitive decline. Given that there are at least thousands to millions of AD patients worldwide who will qualify for the approved therapies, categorization using this evidence-based approach would be key to managing drug access.

Strengths of this study include the multicentric evaluation of blood BD-tau associations with CSF Aβ42/40, AD meta-ROI, and CSF t-tau, BD-tau and NfL in independent cohorts across the AD continuum. Moreover, plasma BD-tau associations with both cross-sectional and longitudinal cognitive decline were evaluated. Additionally, we examined the capacity of combined plasma p-tau and BD-tau abnormalities to recapitulate the link between the A/N biomarkers and cognitive decline. Importantly, all these assessments were performed using multicentric cohorts to demonstrate validity across sites. There are several limitations for this study including the lack of longitudinal plasma BD-tau data. Additionally, BD-tau levels were not different between A+/N+ and A+/N- in cohort 3. This could be seen as a counterintuitive point for the neurodegeneration aspect. The comparison of BD-tau in CN A+ versus CN A- individuals in cohort 3 was not performed due to the lack of enough CN participants. The cohort 3 Aβ+ group, only $n = 4$ participants were CN and thus no reliable cross-sectional statistical analyses for blood BD-tau associations to AD clinical severity could be performed. However, we have demonstrated the prognostic value of elevated BD-tau in prediction of cognitive decline in both cohorts 1 and 3.

In conclusion, plasma BD-tau provides insights into the hitherto elusive AD-type neurodegeneration process in blood. The concentrations of blood BD-tau reflect the combination of pathological Aβ and neurodegeneration which is detrimental to cognitive decline in AD. Blood BD-tau enables staging of Aβ pathophysiology and the risk of cognitive decline according to the presence of neurodegeneration. Individuals with A+/N+, A+/N-, and A-/N- profiles were at the highest, intermediate, and lowest risk for cognitive decline respectively. Moreover, plasma BD-tau can act together with plasma p-tau – as surrogate blood biomarkers of Aβ and neurodegeneration respectively – to operationalize an A/N biomarker combination system in blood that will prove vital to determining who among the thousands of Aβ-positive clinical populations should be prioritized for prescription of the approved anti-Aβ therapies.

## Methods

The present study was performed with the ethical pertinent ethical approvals. Cohort 1 (DDI) has been approved by the Regional Committees for Medical and Health Research Ethics in Norway (REK: 2013/150). All participants gave a written informed consent before participating in the study. A detailed description of inclusion and exclusion criteria are outlined in Fladby et al., (2017)[45]. Cohort 2 (UGOT) was approved by the ethics committees at the University of Gothenburg (#EPN140811). Local ethics committee approval was obtained for cohort 3 (BIODEGMAR). Cohort 4 (MESA) was approved by the University of Washington and Drexel University Institutional Review Boards (protocols #00009029 and #00014523, and #180900605).

### Cohort 1: dementia disease initiation
The Norwegian multi-center study DDI cohort comprise non-demented participants aged between 40 and 80 years with a native

language of either Norwegian, Danish or Swedish. Participants were primarily recruited from memory clinics and advertisements in local news media, recruited at university hospitals across Norway between 2013 - 2022. The cohort comprises CN participants with or without experience of SCD, and patients diagnosed with MCI. For the purposes of the current study, we used the CSF Aβ42/40 ratio to determine Aβ pathology (A+/-, ≤ 0.077), and CSF t-tau to determine neurodegeneration status (N+/-, ≥ 378 pg/mL). We selected a total of $n = 364$ participants with A-/N- status with normal cognition ($n = 157$), A+/N- regardless of clinical status ($n = 45$, CN/MCI, Table 1) and A+/N+ ($n = 162$, CN/MCI Table 1). CN participants were recruited from spouses of patients with dementia/cognitive disorder, and patients who completed lumbar puncture for orthopedic surgery. MCI was diagnosed according to the NIA-AA criteria, which require the presence of self or informant-reported cognitive impairment/decline in combination with lower performance than expected in one or more cognitive domains, yet preserved independence in functional ability and not fulfilling the criteria of dementia. Here, cognitive performance was assessed using a brief neuropsychological battery and deemed CN when performing within the expected normal range (above >−1.5 SD) on all cognitive domains or MCI if having one or more impaired scores (≤-1.5 SD).

The imaging data in cohort-1 was obtained from six sites, across fifteen 1.5 Tesla and four 3.0 Tesla scanner systems, and T1-weighted scans were used for volumetric analysis. We used FreeSurfer (v6.0), an open-source software for neuroimaging analysis for estimation of subregional cortical thickness (https://surfer.nmr.mgh.harvard.edu)[46]. For longitudinal cases, we used the longitudinal stream of FreeSurfer[47]. Subregional cortical thickness was averaged across hemispheres. The AD-meta ROI was determined by creating an average composite thickness of the entorhinal, inferior temporal, middle temporal and the fusiform cortices, as described by Jack et al.[48]. *Cohort 2: University of Gothenburg (UGOT)*

The UGOT cohort ($n = 37$) included paired plasma samples from neurochemically defined AD patients ($n = 28$) and age-matched controls ($n = 9$) from the Sahlgrenska University Hospital, Gothenburg, Sweden. The AD patients were selected based on their core CSF biomarker profile (CSF Aβ42 < 530 pg/ml, CSF p-tau > 60 pg/ml, and CSF t-tau > 350 pg/ml)[27] and had no evidence of other neurological conditions based on routine clinical and laboratory assessments. The control group consisted of selected patients without an AD profile by clinical evaluation and CSF biomarkers.

## Cohort 3: biodegmar memory clinic

The BIODEGMAR cohort is an observational longitudinal study that enrolls patients with neurodegenerative diseases visiting the Cognitive Decline and Movement Disorders Unit of Hospital del Mar Barcelona (Barcelona, Spain). In BIODEGMAR, participants perform a detailed neuropsychological and functional evaluation by a trained neuropsychologist, brain MRI scan, *APOE* ε4 genotyping, lumbar puncture for CSF collection and blood sampling. Clinical evaluation and diagnoses were performed by a specialized neurologist in cognitive disorders. Once a year, BIODEGMAR participants performed a follow-up visit that includes neuropsychological and clinical evaluation. All the included participants in this study presented a Global Deterioration Score (GDS) > 1[49]. The cohort includes CN participants with SCD as well as MCI or dementia patients from different etiologies. Aβ pathology was determined using the CSF Aβ42/40 ratio (A+/- < 0.062) while we used CSF t-tau to determine neurodegeneration status (N+/-, ≥522 pg/mL). A total of $n = 370$ CN, MCI and dementia participants were classified as A-/N- ($n = 111$), A+/N- ($n = 79$) and A+/N + ($n = 180$) (Table 1). A comprehensive description of the BIO-DEGMAR cohort, the inclusion and exclusion criteria, the core AD CSF biomarkers measurements and cutoffs determination was previously published[50–52].

## Cohort 4: MESA-MIND pilot study

The MESA study is comprised of 6814 adults aged 45-84 years free from clinical CVD at baseline (2000-2002) who self-reported their race and ethnicity as White, Black, Hispanic, or Chinese[4]. MESA participants were recruited from six areas in the United States: Forsyth County, North Carolina with balanced recruitment of at least two racial or ethnic groups. At the Forsyth County site, 50% of recruited participants self-reported Black/African American race and 50% were White race. At the 6th examination (2016-2019), all living Forsyth County site participants were recruited to complete detailed cognitive testing, imaging, and blood draw prior to cognitive adjudication as previously described[12].

## CSF collection, biomarker analyses, and *APOE* genotyping
*Cohort 1 and 2*
Lumbar punctures were performed between 9 and 12 AM, and CSF samples were collected in sterile polypropylene tubes and centrifuged. The QuickPlex SQ 120 system from Meso Scale Discovery (MSD, MD, USA) was used to measure Aβ$_{1-42}$ and Aβ$_{1-40}$ in a multiplex setup using the V-plex Ab Peptide Panel 1 (6E10) kit (#K15200E-1). Samples were analyzed in duplicates and reanalyzed if relative deviations (RDs) exceeded 20% and quality control samples with an RD threshold of 15% were controlled for inter-plate and inter-day variation. Commercial enzyme-linked immunosorbent assays (ELISAs) from Innotest, Fujirebio, Ghent, Belgium based on monoclonal antibodies was used to determine CSF concentrations of total tau (t-tau, Tau Ag kits). *APOE* genotyping was performed on EDTA blood samples as previously described[45].

## Cohort 3

Lumbar puncture was performed in the intervertebral space L3/L4, L4/L5, or L5/S1 using a standard needle, between 8 and 11 am. Participants had fasted for at least 8 h. CSF is collected into a 10 ml sterile polypropylene sterile tube (Sarstedt, Nümbrecht, Germany; #62.610.201). Tubes were gently inverted 5–10 times and centrifuged at 2000g for 10 min at 4 °C and aliquoted in volumes of 1.8 ml into sterile polypropylene tubes (1.8 ml cryotube Thermo Scientific Nunc; Thermo Fisher Scientific, Waltham, MA, USA; #377267), and immediately frozen at −80 °C. Blood samples were obtained on the same day as the lumbar puncture and, therefore, in fasting conditions. Whole blood was drawn with a 20 g or 21 g needle gauge into a 10 ml EDTA tubes (BD Vacutainer 10 ml; K2EDTA; #367525). Tubes were gently inverted 5–10 times and centrifuged at 2000g for 10 min at 4 °C. The supernatant was aliquoted in volumes of 1.8 ml into sterile polypropylene tubes (1.8 ml cryotube Thermo Scientific Nunc; Thermo Fisher Scientific, Waltham, MA, USA; #377267), and immediately frozen at −80 °C.

APOE genotyping was performed at Laboratori de Referència de Catalunya (LRC) analysing patients' genomic DNA by means of allelic discrimination's PCR assays using "APOE Real Type" reagents, from Progenie Molecular (Valencia, Spain), studying two polymorphisms: rs7412 (g8041C> T) and rs429358 (g7903 T>C) to define the *APOE* diplotypes for the ε2, ε3 and ε4 alleles, which in turn encode for the most common *APOE* isoforms in the population: apoE2, apoE3, and apoE4 respectively.

The core AD CSF biomarkers (Aβ42, Aβ40, p-tau181, and t-tau) in the BIODEGMAR cohort were measured at LRC with Lumipulse G600II (Fujirebio). Cut-off values for biomarkers and ratios (Aβ42, Aβ42/Aβ40, p-tau181, t-tau, and Aβ42/p-tau181) were previously defined after comparing measures of a group of CN individuals ($n = 42$) with a group of mild AD dementia patients ($n = 48$). A more detailed description of the cohort and biomarkers measurements and cutoffs can be found in previous publications[50,51].

## Cohort 4

DNA was analyzed for *APOE* genotypes performed on EDTA blood samples as previously described[53]; *APOE* ε4 carriage was defined as the

presence of one or more ε4 allele(s). EDTA plasma was collected and stored at -80°C and shipped for analysis on dry ice. CSF was not collected on MESA participants.

## Plasma biomarker analyses

All biomarkers were measured on the Simoa HD-X platform. CSF and BD-tau was measured according to Gonzalez-Ortiz et al., using TauJ5.H3(Bioventix) as capture antibody and Tau12 (BioLegend, #SIG-39416) as partner antibody[23,25]. Plasma p-tau181 was measured either with a commercial method from Quanterix Inc. (pTau-181 V2 Advantage Kit #103714;) or according to the Karikari et al., method[27] using an anti p-tau181 Tau antibody (Thermo Fisher, catalog number: MN1050) as capture and Tau12 (BioLegend, #SIG-39416) as partner antibody. Plasma T-tau and NfL were measured using Quanterix commercial kits (#101552 and #103670). A dilution factor of four was used for the serum and plasma samples, while a dilution factor of thirty was used for measurements in CSF.

## Statistical analyses

All analyses were performed in R studio (R version 4.2.2). Cross-sectional between-group comparisons of continuous variables with assumed normal distributions were performed with ANOVA (age, years of education and blood biomarkers, and MMSE [cohort 3]) or ANCOVA adjusted for age sex and education (CERAD recall and TMT-B [cohort 1]. Nominal variables sex, diagnostic group, and APOE ε4 genotype were assessed with chi-square tests. In cohorts 1 through 3, blood biomarkers were log-transformed (due to skewness) and z-standardized prior to the cross-sectional analyses. Pearson´s correlations were performed between all blood markers against CSF total-tau in the total sample. In cohort-4, we assessed the associations between the plasma biomarkers (BD-tau and NfL) with demographics (age, sex and self-identified race [non-Hispanic White versus African-American]), comorbidities (Diabetes mellitus), risk factors (LDL, HDL, total cholesterol, Body Mass Index, systolic and diastolic blood pressure), genetic risk (APOE ε4) and renal function (eGFR). In cohort-1, associations between plasma BD-tau, t-tau and NfL with age, sex, APOE ε4 and eGFR were assessed. In cohort-3, associations between serum BD-tau, plasma t-tau and plasma NfL with age and sex and APOE ε4 were assessed. In cohort-1, the CN A-/N- was used as the reference group and mean z-scores for A+/N- and A+/N+ group thus reflect standard deviations (SDs) from the reference group. These analyses were repeated for the cohort-2 sample (CN A-/N- vs dementia A+/N+). For cohort-3, the sample was mixed SCD/MCI and dementia in all A/N groups (A-/N- vs A+/N- and A+/N+). In cohort-1, linear mixed models (LMMs) were used to assess longitudinal associations between baseline CSF, plasma biomarkers, raw CERAD memory recall and log-transformed TMT-B completion times (in seconds). The covariates age at baseline, years of education and sex were included for CERAD models and age and years of education were included for TMT-B models. Random slopes for time (years since baseline) were fitted for all models following results from log-likelihood ratios tests, indicating improved model fits. Similarly, LMMs were fitted to measure longitudinal associations between the biomarkers and MRI AD-Meta ROI. Here, the inclusion of random slopes did not change the interpretation of the results and models were thus omitted. The MRI data included in the AD-meta ROI was harmonized by Longitudinal ComBat to deal with technical variability across different scanner systems[54]. LongCombat adjusted MRI data were determined by including age, sex and time*-biomarker for the main longitudinal models[55]. For the plasma-determined A/N models, age, sex and time*A/N groups were included in the model.

In cohort 3, similar models were fitted. Here, with CDR and MMSE as dependent variables, and age, sex, and education included as covariates. Only random intercepts were fitted. For LMMs, all continuous variables, except years since baseline were standardized prior to analysis. In both cohorts 1 and 3, Receiver operating curves (ROCs) were fitted for blood BD-tau and plasma p-tau181 and an optimal cut-off using the Youden index were applied. We used the A-/N- and A+/N+ groups to determine pertinent cut-offs for both markers. We used the resulting cut-offs to determine blood-based A/N groups, treating p-tau181 as a marker of Aβ pathology (A+/-) and BD-tau as a marker of neurodegeneration (N+/-). We then fitted LLMs to assess the longitudinal associations between blood-based A/N groups and cognitive performance (cohort 1 and 3) and AD Meta-ROI atrophy (cohort 1). No adjustment for multiple testing was conducted in any of the models.

## Supplementary statistical analyses and results

For cohort 1, between-group comparisons of CSF BD-tau and NfL are shown in Supplementary Table 6. LLMs for longitudinal associations between baseline CSF BD-tau and NfL with future CERAD memory decline, worsening of TMT-B as well as AD-Meta ROI atrophy are included in Supplementary Table 7.

## Reporting summary

Further information on research design is available in the Nature Portfolio Reporting Summary linked to this article.

## Data availability

Anonymized aggregated level data will be shared by request from a qualified academic investigator for the sole purpose of replicating procedures and results presented in the article, and as long as data transfer is in agreement with EU legislation on the general data protection regulation and decisions by the Ethical Review Boards in charge of each of the cohorts used for this study.

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

## Acknowledgements

The authors would like to express their most sincere gratitude to the cohort participants and relatives without whom this research would not have been possible. For Biodegmar (cohort-3), collaborators of the study include: Isabel Estragues, Oriol Grau-Rivera, Rosa María Manero, Víctor Puente-Periz, Jaume Roquer. We also would like to thank Paula Ortiz-Romero for technical support, and all the staff at the Department of Neurology at Hospital del Mar who supported this project. The project was funded by the Norwegian Research Council, JPND/PMI-AD (NRC 311993), the National Institute of Health (R01 AG083874-01). F.G.-O. was funded by the Anna Lisa and Brother Björnsson's Foundation and Emil och Maria Palms Foundation. MSC receives funding from the European Research Council (ERC) under the European Union's Horizon 2020 research and innovation programme (Grant agreement No. 948677), Project "PI19/00155", funded by Instituto de Salud Carlos III (ISCIII) and co-funded by the European Union, and from a fellowship from"la Caixa" Foundation (ID 100010434) and from the European Union's Horizon 2020 research and innovation programme under the Marie Skłodowska-Curie grant agreement No 847648 (LCF/BQ/PR21/11840004). TKK was funded by the NIH (R01 AG083874-01, U24 AG082930-01, P30 AG066468, RF1 AG052525-01A1, R01 AG053952-05, R37 AG023651-17, RF1 AG025516-12A1, R01 AG073267-02, R01 AG075336-01, R01 AG072641-02, and P01 AG025204-16), the Swedish Research Council (Vetenskåpradet; 2021-03244), the Alzheimer's Association (AARF-21-850325), the Swedish Alzheimer's Foundation (Alzheimerfonden), Hjärnfonden (FO2021-0298) the Aina (Ann) Wallströms and Mary-Ann Sjöbloms stiftelsen, and the Emil och Wera Cornells stiftelsen. K.B. is supported by the Swedish Research Council (#2017-00915 and #2022-00732), the Swedish Alzheimer Foundation (#AF-930351, #AF-939721 and #AF-968270), Hjärnfonden, Sweden (#FO2017-0243 and #ALZ2022-0006), the Swedish state under the agreement between the Swedish government and the County Councils, the ALF-agreement (#ALFGBG-715986 and #ALFGBG-965240), the Alzheimer's Association 2021 Zenith Award (ZEN-21-848495), and the Alzheimer's Association 2022-2025 Grant (SG-23-1038904 QC). ING receives funding from the Spanish Institute of Health Carlos III by project reference AC20/00001, P.I. PI21/00194. B.E.K. was supported by a grant from Helse-Nord (HNF1540-20).

## Author contributions

F.G.-O., B.E.K., T.K.K., T.F. and K.B. designed the study. F.G.-O., A.H. P.R.K., and J.N. acquired the blood biomarker data. Data analysis was performed by B.E.K. and J.C.; J.E.T. and S.T. Clinical data was acquired by P.S., B.G., L.P., M.S.H., J.J., R.S., G.B., G.G., A.B., K.W., D.A., A.F.-L, G.G.-E., I.N.-G., J.L., and K.M.H. Reagents for the BD-tau assay were provided by M.T. and P.H. The original draft of the manuscript was written by F.G.-O., B.E.-K., and T.K.K. Critical feedback for the latest version of the manuscript was provided by H.Z., T.M.H., A.P.P., M.S.C., T.F. and K.B. All authors reviewed the final version of the manuscript.

## Funding

## Competing interests

M.T. and P.H. are employees of Bioventix Plc. H.Z. has served at scientific advisory boards and/or as a consultant for Abbvie, Alector, Annexon, Artery Therapeutics, AZTherapies, CogRx, Denali, Eisai, Nervgen, Pinteon Therapeutics, Red Abbey Labs, Passage Bio, Roche, Samumed, Siemens Healthineers, Triplet Therapeutics, and Wave, and has given lectures in symposia sponsored by Cellectricon, Fujirebio, Alzecure, Biogen, and Roche. K.B. has served as a consultant and at advisory boards for Acumen, ALZPath, BioArctic, Biogen, Eisai, Lilly, Moleac Pte. Ltd, Novartis, Ono Pharma, Prothena, Roche Diagnostics, and Siemens Healthineers; has served at data monitoring committees for Julius Clinical and Novartis; has given lectures, produced educational materials and participated in educational programs for AC Immune, Biogen, BioArctic, Celdara Medical, Eisai and Roche Diagnostics. H.Z. and K.B. are co-founders of Brain Biomarker Solutions in Gothenburg AB, a GU Ventures-based platform company at the University of Gothenburg. M.S.C. has served as a consultant and at advisory boards for Roche Diagnostics International Ltd and Grifols S.L., has given lectures in symposia sponsored by Roche Diagnostics, S.L.U. and Roche Farma, S.A., and was granted with a project funded by Roche Diagnostics International Ltd; payments were made to the institution (BBRC). A.P.P. has served on advisory boards for Schwabe Farma Iberica. T.K.K. has received honoraria from the University of Wisconsin Madison and the University of Pennsylvania and has an awarded patent (#WO2020193500A1), all unrelated to this work. B.E.K. has served as a consultant for Biogen and advisory board for Eisai. T.F. has served as a consultant and at the advisory boards for Biogen, Novo Nordisk, Eli Lilly, and Roche. The other authors declare no competing interest.

## Additional information

[1]Department of Psychiatry and Neurochemistry, Institute of Neuroscience and Physiology, the Sahlgrenska Academy at the University of Gothenburg, Mölndal, Sweden. [2]Clinical Neurochemistry Laboratory, Sahlgrenska University Hospital, Mölndal, Sweden. [3]Department of Neurology, University Hospital of North Norway, Tromsø, Norway. [4]Department of Psychology, Faculty of Health Sciences, The Arctic University of Norway, Tromsø, Norway. [5]Institute of Clinical Medicine, Campus Ahus, University of Oslo, Oslo, Norway. [6]Barcelonaβeta Brain Research Center (BBRC), Pasqual Maragall Foundation, Barcelona, Spain. [7]Hospital del Mar Research Institute, Barcelona, Spain. [8]Cognitive Decline and Movement Disorders Unit, Neurology Department, Hospital del Mar, Barcelona, Spain. [9]Department of Internal Medicine, Section on Gerontology and Geriatric Medicine, Wake Forest University School of Medicine, Winston-Salem, NC, USA. [10]Department of Neurology, Akershus University Hospital, Lørenskog, Norway. [11]Research Group for Age-Related Medicine, Haugesund Hospital, Haugesund, Norway. [12]Department of Neuropsychology, Haugesund Hospital, Haugesund, Norway. [13]Department of Clinical Medicine (K1), University of Bergen, Bergen, Norway. [14]Department of Geriatric Medicine, Haraldsplass Deaconess Hospital, Bergen, Norway. [15]Department of Clinical Sciences, Faculty of Medicine, University of Bergen, Bergen, Norway. [16]Department of Neurology and Clinical Neurophysiology, University Hospital of Trondheim, Trondheim, Norway. [17]Department of Neuromedicine and Movement Science, Faculty of Medicine and Health Sciences, Norwegian University of Science and Technology, Trondheim, Norway. [18]Department of Physics, University of Oslo, Oslo, Norway. [19]Unit for Computational Radiology and Artificial Intelligence, Oslo University hospital, Oslo, Norway. [20]Department of Psychology, Faculty for Social Sciences, University of Oslo, Oslo, Norway. [21]Department of Old Age Psychiatry. Institute of psychiatry, Psychology and Neuroscience King's College London, London, UK. [22]Centre for Age-Related Diseases, University Hospital Stavanger, Stavanger, Norway. [23]Department of Medicine and Life Sciences, Universitat Pompeu Fabra, Barcelona 08003, Spain. [24]ERA-Net on Cardiovascular Diseases (ERA-CVD) consortium, Barcelona, Spain. [25]Bioventix Plc, 7 Romans Business Park, East Street, Farnham, Surrey GU9 7SX, UK. [26]Department of Medicine, Universitat Autònoma de Barcelona, Barcelona, Spain. [27]Department of Neurodegenerative Disease, UCL Institute of Neurology, Queen Square, London, UK. [28]UK Dementia Research Institute at UCL, London, UK. [29]Hong Kong Center for Neurodegenerative Diseases, Clear Water Bay, Hong Kong, China. [30]Wisconsin Alzheimer's Disease Research Center, University of Wisconsin School of Medicine and Public Health, University of Wisconsin-Madison, Madison, WI, USA. [31]Centro de Investigación Biomédica en Red de Fragilidad y Envejecimiento Saludable (CIBERFES), Madrid, Spain. [32]Department of Psychiatry, University of Pittsburgh, Pittsburgh, PA, USA. [33]These authors contributed equally: Fernando Gonzalez-Ortiz, Bjørn-Eivind Kirsebom. [34]These authors jointly supervised this work: Thomas K. Karikari; Tormod Fladby, Kaj Blennow. ✉e-mail: Fernando.gonzalez.ortiz@gu.se

