## [Peer Review File · Nature Communications]

Plasma brain-derived tau is an amyloid-associated neurodegeneration biomarker in Alzheimer's diseaseREVIEWER COMMENTS

Reviewer #1 (Remarks to the Author):

The manuscript describes the possible applicability of brain-derived tau (BD-tau) as a blood-based measure of AD-related neurodegeneration. The foundation of the work is potentially impactful as existing blood-based measures of neurodegeneration (such as NfL) are very nonspecific. The manuscript is clearly and thoughtfully written and the data analyses are appropriate, making the manuscript certainly worth publishing where it will add to knowledge in the field. There are some limitations noted below which I suspect could dampen the impact.

1. It is not clear from the presented results how BD-tau performs in cognitively impaired individuals who are amyloid negative. If BD-tau is elevated in that group, it would argue against the conclusion that this marker is specifically AD-relevant.
2. In one of the two major cohorts, BD-tau levels did not appear different in A+/N+ versus A+/N-. This also seems to cut against some of the claims in the abstract about the strength of findings.
3. The specific plasma biomarkers used to determine amyloid status are also a limitation. The assays appear to all be via ECL rather than mass spectrometry (the latter being more widely recognized as having markedly better performance). The use of AB42/40 in one cohort and AB42 in the other is a mismatch and at least raises the question of how strong these results are. The lack of use of a p-tau isoform (as the authors note, p-tau 217 and p-tau 231 proving in many studies to be the most robust blood measures of amyloidosis; even p-tau 181 having a strong role there). This is also against the backdrop that amyloid PET is not utilized as comparator.
4. The description of the study involving 4 cohorts is technically correct but I would suggest modulating the language (particularly in the abstract) on this front, as the analyses were fairly delineated (i.e., blood vs. blood/CSF biomarkers in 1-2 cohorts, blood vs. MRI separately, blood vs. cognition separately).
5. The cognitive associations appear quite modest and limited to very granular measures of CERAD memory subscale and TMT-B. With the reported data, it is hard to place in context whether there really is a difference in BD-tau versus NfL for these outcomes, or whether these reflect a degree of variation across multiple measures for multiple outcomes (each of which is having a modest strength of relationship). Also, did the authors have access to and/or analyze other cognitive and functional measures (I would have to assume these are available in the primary dataset) to provide further context?

Reviewer #2 (Remarks to the Author):

In this manuscript, Gonzalez-Ortiz et al explore the associations of a novel biomarker, brain-derived tau in plasma, with other markers and its potential as a marker for amyloid-associated neurodegeneration in Alzheimer's disease (AD). In addition to the novelty of the marker, the paper has remarkable strong points, such as the inclusion of four cohorts with a large number of participants across different stages of

the AD continuum and its comprehensive investigation into associations with various fluid, imaging, and cognitive measures.

To improve the clarity of the manuscript I suggest addressing the following points:

1. Throughout the manuscript and in the abstract, the authors refer to BD-tau as a marker of “AD-type neurodegeneration”. However, they found higher concentration of blood BD-tau in A+N+ groups, but also in the A+N- group in cohort 3. If BD-tau is a marker of neurodegeneration, how do the authors explain the elevation in the A+N- group?
2. The study did not include markers of the T category. To what extent can the authors be certain that BD-tau aligns more with the N category rather than the T category? It would be interesting to have information about the association of BD-tau with CSF p-tau.
3. In the results section (“Combining plasma p-tau and BD-tau operationalizes the A/N classification in blood”), the authors evaluated the use of p-tau and BD-tau as A and N markers, respectively. But to do so, they use a combination of CSF biomarkers (A-/N- and A+/N+) as the standard of truth to calculate Youden-based cutoffs for both markers. Why was this approach taken instead of applying independent categories as the standard for each marker (CSF A+ and CSF A- as the standard in the A category for p-tau ; CSF N+ and CSF N- as the standard in the N category for BD-tau)?
4. BD-tau was measured in plasma in cohorts 1 and 4 and in serum in cohorts 2 and 3. The authors have previously shown that measures in both matrices have equivalent diagnostic performance. However, as the results study and compare its association with other markers, it would be useful to have some information on the correlation (and whether it is lineal) between measures in serum and plasma samples.
5. The presentation of the detailed comparisons, associations and correlations in the four cohorts is difficult to follow in the main text, making it challenging for readers to synthesize the findings effectively. To improve the paper's readability, it would be beneficial to restructure the results section to summarize the main findings within the text leaving most of the numeric details for tables and figures. This would facilitate a clearer and more accessible presentation of the results.

Reviewer #3 (Remarks to the Author):

Authors have already reported the “blood-based Brain-derived-tau (BD-tau)” in previous paper “Gonzalez-Ortiz F et al. Brain. Published online December 27, 2022:awac407. doi:10.1093/brain/awac407”. Authors analyzed the BD-tau in large-scale clinical cohorts to further characterize the biomarker for establishing the clinical utility. To discuss the use of this biomarker in clinical settings, I suggest to update the following information in the manuscript;

1. Authors hypothesized "plasma BD-tau will increase according to A β - pathophysiology if it is an AD-associated neurodegeneration marker. To investigate this, authors grouped participants in cohorts 1 through 3 according to joint A and/or N abnormalities (A+/N- or A+/N+), as compared with CU participants with normal biomarkers (e.g., A-/N-). Authors used CSF A β 42/40 ratio (cohort-1), A β 42 (cohorts 1 & 2) and t-tau (cohorts 1-3) as the standard A and N biomarkers because, similar to their plasma alternatives, they reflect soluble biomarker changes that tend to become abnormal earlier in AD than neuroimaging biomarkers."

> For this point, I suggest to use not only "t-tau" but also other representative N biomarkers such as "neurogranin" and "neurofilament light" to define N- or N+. Total tau (t-tau) is regarded as the controversial biomarker with double characteristics recapitulating the amyloid pathology as well as neurodegeneration. The previous report from Sato et al. (Neuron 2018, doi: 10.1016/j.neuron.2018.02.015.) clarified that the total-tau would actively (not passively) be secreted to the extracellular space including CSF by responding to the amyloid pathology in early AD stages. Since the increase of t-tau in early stage of AD would be highly contributed by amyloid pathology only, the use of only t-tau as "N-marker signature" in CN and MCI stages will not be appropriate in Figure 1. By adding other cases with "neurogranin" and "neurofilament light" as N classifiers, we will further understand the characteristic/how to use of blood BD-tau.

2. Authors mentioned "Even though a strong correlation between CSF t-tau and CSF BD-tau ($r=0.91$, $p<.001$; Extended Data Figure 2A) across A/N groups was found, the correlation of plasma BD-tau with CSF t-tau in cohort-1 was more modest ($r=0.42$, $p<.001$; Extended Data Figure 3A), but comparatively stronger than for plasma t-tau with CSF t-tau ($r=0.23$, $p<.001$; Extended Data Figure 3B) and plasma NfL with CSF t-tau ($r=0.33$, $p<.001$; Extended Data Figure 3C). Importantly, the strength of the correlation between plasma BD-tau with CSF t-tau tended to increase according to concomitant A/N positivity but only the A+N+ group showed significance."

> For this point, authors should make the extensive discussion to explain the potential reason to see these results and suggest the mitigation plan to improve the assay. Why BD-tau has very modest translatability in blood even though the authors clearly mention that the species are from brain CNS completely?

3.

> Although the authors showed some information that combining plasma p-tau and BD-tau as blood biomarkers of A and N respectively provides a novel approach to stage the severity of Amyloid-pathophysiology and the risk for near-term cognitive decline across the AD continuum, I am still wondering if the current plasma BD-tau assay offers the additional benefits for the cognitive status (as a phenotype of neurodegeneration) upon the other AD biomarkers including p-tau species in "clinical settings". To further highlight the clinical utility of blood BD-tau, I suggest to do e.g., ANCOVA of blood BD-tau levels to see the levels were elevated in cognitively impaired participants compared to cognitively normal participants, even after adjustment for age, sex, CSF A β 42/40, CSF (or blood) p-tau181, and CSF (or blood) NfL levels. Also, authors should investigate if the BD-tau levels statistically-significantly improve prediction of cognitive status even after adjusting for covariates and other major AD CSF

biomarkers (age, sex, CSF A β 42/40, CSF p-tau181, and CSF NfL levels) by the appropriate logistic regression model of cognitive status (Cognitively-unimpaired versus Cognitively-impaired) as a function of blood BD-tau.

REVIEWER COMMENTS

Reviewer #1 (Remarks to the Author):

The manuscript describes the possible applicability of brain-derived tau (BD-tau) as a blood-based measure of AD-related neurodegeneration. The foundation of the work is potentially impactful as existing blood-based measures of neurodegeneration (such as NfL) are very nonspecific. The manuscript is clearly and thoughtfully written, and the data analyses are appropriate, making the manuscript certainly worth publishing where it will add to knowledge in the field. There are some limitations noted below which I suspect could dampen the impact.

1.- It is not clear from the presented results how BD-tau performs in cognitively impaired individuals who are amyloid negative. If BD-tau is elevated in that group, it would argue against the conclusion that this marker is specifically AD-relevant.

Author reply: *Thank you for this comment. In the DDI cohort, no cognitively impaired (CI) cases with A-/N+ or A-/N- were included. For this reason, such an analyses could not be carried out. In the Biodegmar cohort, which was from a memory clinic, the majority of cases were cognitively impaired (MCI or dementia) but others had SCD (thus classified as cognitively unimpaired, CN). We have now performed an analysis to assess differences in serum BD-tau concentrations between the CN and CI participants with an A-/N- profile. The results showed non-significant differences between these group,s indicating that the our findings were independent of cognitive status.*

Moreover, there were no cross-sectional or longitudinal differences in the CDR and the MMSE scores and trajectories between the A-/N+ & A-/N- groups determined using serum p-tau and BD-tau (Figure 4 D, E and Extended Data Table 5). Similarly, we found no difference in the concentrations of serum BD-tau at baseline in the A-/N+ & A-/N- groups (Figure or Table X). To the contrary, the A+ participants (that is, the A+/N- and A+/N+ patients) all showed worse performances on CDR with time compared with the A-/N- group. These results are consistent with those shown for the cognitively unimpaired participants in the DDI cohort, meaning our findings that BD-tau levels and biomarker performances are driven by amyloid positivity hold true irrespective of the cognitive capabilities of the participants being evaluated.

2.- In one of the two major cohorts, BD-tau levels did not appear different in A+/N+ versus A+/N-. This also seems to cut against some of the claims in the abstract about the strength of findings.

Author reply: *We thank the reviewer for the opportunity to further clarify this point. We have shown that the amyloid- and neurodegeneration-dependent increases of blood BD-tau may be additionally modulated by the severity of cognitive impairment. This is explainable by the observation that contrary to the mostly cognitively unimpaired DDI cohort where there were clearly higher levels of BD-tau in the A+/N+ versus A+/N- groups, the differences in the memory clinic-based Biodegmar cohort was only marginal and thus not statistically significant. Importantly, it is not only BD-tau levels that did not differ between these symptomatic groups; cognitive performance, evaluated with both the MMSE and the CDR, likewise did not differ between the two groups. Since the two groups consisted of both MCI and dementia patients, a reasonable next step would be to do sub-analysis in these clinical groups separately. However, that would reduce the statistical power to draw any strong conclusion. Moreover, the lack of MRI data for the Biodegmar cohort meant we could not further evaluate if structural MRI would provide useful insights. We have added a Discussion point addressing.*

3.- The specific plasma biomarkers used to determine amyloid status are also a limitation. The

assays appear to all be via ECL rather than mass spectrometry (the latter being more widely recognized as having markedly better performance). The use of AB42/40 in one cohort and AB42 in the other is a mismatch and at least raises the question of how strong these results are. The lack of use of a p-tau isoform (as the authors note, p-tau 217 and p-tau 231 proving in many studies to be the most robust blood measures of amyloidosis; even p-tau 181 having a strong role there). This is also against the backdrop that amyloid PET is not utilized as comparator.

Author reply: Thank you for the opportunity to clarify this point as it may have been confusing to the reviewer. We did not measure AB peptides in blood, but only in CSF. Both the DDI and Biodegmar cohorts used CSF AB42/40 for amyloid status determination, and only the UGOT cohort used CSF AB42. However, the UGOT cohort comprised only of healthy controls and clinically adjudicated AD dementia, with a wide margin of amyloid pathology between them. Thus, we argue that this is not a great source of error in our analyses.

The CSF AB42/40 assay used for Biodegmar was the FDA-approved method on the Lumipulse platform, with validated cut points. Measuring AB peptides in CSF does not require mass spectrometry. In fact, previous assay comparison studies have reported exceptionally strong correlation (>0.9) and performance interchangeability between immunoassay and LC-MS/MS AB assays measured in CSF (see for example, Shaw et al., 2019 *Clinical Biochemistry*; <https://www.sciencedirect.com/science/article/pii/S0009912019304886>). As referenced in the Methods section (<https://pubmed.ncbi.nlm.nih.gov/34645914/>) and in the absence of a validated value, the Mesoscale AB42/40 cut-off used for the DDI cohort was determined using ROC analyses with amyloid PET as the standard of truth, yielding an AUC of .957 (95% CI: 903-1).

On the use of CSF AB42/40 and not amyloid PET, there is very strong correlation and agreement between the two modalities replicated across multiple independent studies (e.g., <https://n.neurology.org/content/85/14/1240> and see our recent reviews including PMID 35585226 and 36927491). Moreover, the CSF AB42/40 and AB42 appear to become abnormal ahead of AB PET since the former measure soluble biochemical forms while the latter requires multimeric aggregates to give a reliable signal (e.g., <https://academic.oup.com/brain/article/139/4/1226/2464326>). In this study, we were mostly interested in biofluid biomarkers, therefore CSF AB42 or AB42/40 was preferable.

Finally, plasma p-tau181 was the biomarker readily available for the cohorts. Future work will evaluate p-tau217 and p-tau231 in the proposed blood-based A/N framework.

4.- The description of the study involving 4 cohorts is technically correct, but I would suggest modulating the language (particularly in the abstract) on this front, as the analyses were fairly delineated (i.e., blood vs. blood/CSF biomarkers in 1-2 cohorts, blood vs. MRI separately, blood vs. cognition separately).

Author reply: We have updated the text based on the comments of the reviewers to facilitate a more accessible reading and interpretation of the results.

5.- The cognitive associations appear quite modest and limited to very granular measures of CERAD memory subscale and TMT-B. With the reported data, it is hard to place in context whether there really is a difference in BD-tau versus NfL for these outcomes, or whether these reflect a degree of variation across multiple measures for multiple outcomes (each of which is having a modest strength of relationship). Also, did the authors have access to and/or analyze other cognitive and functional measures (I would have to assume these are available in the primary dataset) to provide further context?

Author reply: Thank you. Indeed, the associations are quite modest in cohort 1 due to the participants including a large number of cognitively unimpaired individuals. DDI is also younger than most AD/ADRD cohorts (~63 years on average in the DDI cohort) and the few MCI cases included were largely in the “early MCI” phase. Additionally, we have previously shown that this cohort has a fairly slow progression to dementia (<https://pubmed.ncbi.nlm.nih.gov/36262371/>). Thus, a granular measure, such as the CERAD memory or TMT-B is needed to ascertain cognitive changes. While CDR and MMSE is available in the DDI cohort – these measures are indeed too coarse for use in this cohort. We have provided further interpretation in the revised manuscript.

Reviewer #2 (Remarks to the Author):

In this manuscript, Gonzalez-Ortiz et al explore the associations of a novel biomarker, brain-derived tau in plasma, with other markers and its potential as a marker for amyloid-associated neurodegeneration in Alzheimer's disease (AD). In addition to the novelty of the marker, the paper has remarkable strong points, such as the inclusion of four cohorts with a large number of participants across different stages of the AD continuum and its comprehensive investigation into associations with various fluid, imaging, and cognitive measures. To improve the clarity of the manuscript I suggest addressing the following points:

1.- Throughout the manuscript and in the abstract, the authors refer to BD-tau as a marker of “AD-type neurodegeneration”. However, they found higher concentration of blood BD-tau in A+N+ groups, but also in the A+N- group in cohort 3. If BD-tau is a marker of neurodegeneration, how do the authors explain the elevation in the A+N- group?

Author reply: Thank you, we have addressed this in point 2 above for reviewer 1.

2.- The study did not include markers of the T category. To what extent can the authors be certain that BD-tau aligns more with the N category rather than the T category? It would be interesting to have information about the association of BD-tau with CSF p-tau.

Author reply: In our previous study (Gonzalez-Ortiz et al., 2023 Brain), we showed that while BD-tau correlates with CSF p-tau (measure of “T”), there is a much stronger correlation with CSF t-tau (measure of “N”). These findings are further extended in the current manuscript where the correlation of plasma BD-tau with CSF t-tau is strong across all the A/N categories. Moreover, and as explained in the Introduction and Discussion sections as well as in the 2011 update of the NIA-AA framework, the presence of neurodegeneration is thought to potentiate amyloid effects hence our hypothesis.

3.- In the results section (“Combining plasma p-tau and BD-tau operationalizes the A/N classification in blood”), the authors evaluated the use of p-tau and BD-tau as A and N markers, respectively. But to do so, they use a combination of CSF biomarkers (A-/N- and A+/N+) as the standard of truth to calculate Youden-based cutoffs for both markers. Why was this approach taken instead of applying independent categories as the standard for each marker (CSF A+ and CSF A- as the standard in the A category for p-tau ; CSF N+ and CSF N- as the standard in the N category for BD-tau)?

Author reply: In the present work, we only included FDA-approved and widely used CSF biomarker-determined A-/N-, A+/N- and A+/N+ groups from all cohorts. Thus, the A+/N+ cases were the only available cases with N+ to carry out ROC analyses and derive cut-offs for plasma BD-tau. To prevent introducing a bias towards A+-driven neurodegeneration in our analyses, we opted to run a ROC in the entire DDI cohort with CSF t-tau-derived N+ and N- (cohort-1,

n=648, of which 270 were N+ and 378 N-) and compare the AUCs and cut-offs produced. This produced a lower, yet similar AUC (.664, 95%CI: .622-.706) as in the subsample selected for this work (.700, 95%CI: .638-.761). Moreover, the cut-offs were also similar (≥ 5.77 and ≥ 5.31 respectively). Furthermore, we applied the alternate cut-off at 5.77 to our longitudinal models for cognition (TMTB and CERAD recall), and they yielded the same conclusions as with the 5.31 cut-off (data not shown).

4.- BD-tau was measured in plasma in cohorts 1 and 4 and in serum in cohorts 2 and 3. The authors have previously shown that measures in both matrices have equivalent diagnostic performance. However, as the results study and compare its association with other markers, it would be useful to have some information on the correlation (and whether it is lineal) between measures in serum and plasma samples.

Author reply: *Thank you, this is a good point. However, we have previously reported the comparison of plasma and serum BD-tau in two independent cohorts, and this has been referenced in the current work (see methods section). The results show that plasma and serum BD-tau linearly and strongly correlate in AD and non-AD conditions. Reference: <https://doi.org/10.1002/alz.13156>*

5.- The presentation of the detailed comparisons, associations and correlations in the four cohorts is difficult to follow in the main text, making it challenging for readers to synthesize the findings effectively. To improve the paper's readability, it would be beneficial to restructure the results section to summarize the main findings within the text leaving most of the numeric details for tables and figures. This would facilitate a clearer and more accessible presentation of the results.

Author reply: *We have updated the text based on the comments of the reviewers to facilitate a more accessible reading and interpretation of the results*

Reviewer #3 (Remarks to the Author):

Authors have already reported the “blood-based Brain-derived-tau (BD-tau)” in previous paper “Gonzalez-Ortiz F et al. Brain. Published online December 27, 2022:awac407. doi:10.1093/brain/awac407”. Authors analyzed the BD-tau in large-scale clinical cohorts to further characterize the biomarker for establishing the clinical utility. To discuss the use of this biomarker in clinical settings, I suggest to update the following information in the manuscript;

1.- Authors hypothesized "plasma BD-tau will increase according to A β pathophysiology if it is an AD-associated neurodegeneration marker. To investigate this, authors grouped participants in cohorts 1 through 3 according to joint A and/or N abnormalities (A+/N- or A+/N+), as compared with CU participants with normal biomarkers (e.g., A-/N-). Authors used CSF A β 42/40 ratio (cohort-1), A β 42 (cohorts 1 & 2) and t-tau (cohorts 1-3) as the standard A and N biomarkers because, similar to their plasma alternatives, they reflect soluble biomarker changes that tend to become abnormal earlier in AD than neuroimaging biomarkers."

> For this point, I suggest to use not only “t-tau” but also other representative N biomarkers such as “neurogranin” and “neurofilament light” to define N- or N+. Total tau (t-tau) is regarded as the controversial biomarker with double characteristics recapitulating the amyloid pathology as well as neurodegeneration. The previous report from Sato et al. (Neuron 2018, doi: 10.1016/j.neuron.2018.02.015.) clarified that the total-tau would actively (not passively) be secreted to the extracellular space including CSF by responding to the amyloid pathology in early AD stages. Since the increase of t-tau in early stage of AD would be highly contributed

by amyloid pathology only, the use of only t-tau as “N-marker signature” in CN and MCI stages will not be appropriate in Figure 1. By adding other cases with “neurogranin” and “neurofilament light” as N classifiers, we will further understand the characteristic/how to use of blood BD-tau.

Author reply: We thank the reviewer for this point. However, neurogranin was not available for all the cohorts. Moreover, please see our previous work on neurogranin in the AD A/T/N continuum (<https://pubmed.ncbi.nlm.nih.gov/36262371/>) showing that only A+/N+ and not A+/N- cases had elevated neurogranin concentrations (and thus also normal t-tau). Neurogranin and t-tau were highly correlated in and thus we argue that neurogranin is likely not an independent “N” marker. As for NfL, in Extended Data Figure 1, we have shown the equivalent of Figure 1 using plasma t-tau or NfL instead of BD-tau and have described and discussed the results in the text, indicating that the others do not show the same strong amyloid-associated increases in the AD continuum as BD-tau does. However, as our results show, while BD-tau in blood seems to be slightly better than NfL in predicting future cognitive decline and MRI AD meta-ROI atrophy in early-stage cases (in the DDI cohort), the differences between the markers are not large. This may be due to t-tau elevation being more AD specific than NfL, and thus more sensitive in early stages of the AD clinical continuum. In Biodegmar, less of a distinction is seen – which may point to NfL as a later marker owing to more widespread pathology (i.e. likely more frequent comorbidities). Thus, in future work, we aim to combine “N” markers in blood in the AD continuum, as these likely address different aspects of neurodegeneration and followingly, cognitive impairment and decline.

2.- Authors mentioned "Even though a strong correlation between CSF t-tau and CSF BD-tau ($r=0.91$, $p<.001$; Extended Data Figure 2A) across A/N groups was found, the correlation of plasma BD-tau with CSF t-tau in cohort-1 was more modest ($r=0.42$, $p<.001$; Extended Data Figure 3A), but comparatively stronger than for plasma t-tau with CSF t-tau ($r=0.23$, $p<.001$; Extended Data Figure 3B) and plasma NfL with CSF t-tau ($r=0.33$, $p<.001$; Extended Data Figure 3C). Importantly, the strength of the correlation between plasma BD-tau with CSF t-tau tended to increase according to concomitant A/N positivity but only the A+N+ group showed significance."

> For this point, authors should make the extensive discussion to explain the potential reason to see these results and suggest the mitigation plan to improve the assay. Why BD-tau has very modest translatability in blood even though the authors clearly mention that the species are from brain CNS completely?

Author reply: Indeed, correlations between CSF and blood biomarkers are moderate. However, this is generally the case in most studies, not just in ours. Here, the correlation between CSF and plasma markers appears to be influenced not only by the origin of the protein measured but also by different factors including the clearance factors in blood. Thus, even with a very CNS specific marker (such as BD-tau) we don't expect to see a close to perfect correlation between CSF and plasma for such a marker. If we assume that the p-tau markers are CNS specific, the correlation plasma-CSF BD-tau that we observed in this study is similar - and in some cases stronger - to the correlation between plasma and CSF p-tau markers reported using different assays in the same cohort – Biodegmar-
<https://dx.doi.org/10.1002/alz.12841>.
[Microsoft Word - alz12841-sup-0001-SupMat.docx \(wiley.com\)](https://www.wiley.com/doi/suppl/10.1002/alz.12841/suppl-0001-sup-0001-SupMat.docx)

3.- Although the authors showed some information that combining plasma p-tau and BD-tau as blood biomarkers of A and N respectively provides a novel approach to stage the severity of Amyloid- pathology and the risk for near-term cognitive decline across the AD continuum, I am still wondering if the current plasma BD-tau assay offers the additional benefits

for the cognitive status (as a phenotype of neurodegeneration) upon the other AD biomarkers including p-tau species in “clinical settings”. To further highlight the clinical utility of blood BD-tau, I suggest to do e.g., ANCOVA of blood BD-tau levels to see the levels were elevated in cognitively impaired participants compared to cognitively normal participants, even after adjustment for age, sex, CSF A β 42/40, CSF (or blood) p-tau181, and CSF (or blood) NfL levels. Also, authors should investigate if the BD-tau levels statistically-significantly improve prediction of cognitive status even after adjusting for covariates and other major AD CSF biomarkers (age, sex, CSF A β 42/40, CSF p-tau181, and CSF NfL levels) by the appropriate logistic regression model of cognitive status (Cognitively unimpaired versus Cognitively-impaired) as a function of blood BD-tau.

Author reply: *Thank you. We have already included this distinction for the DDI cohort (Extended Data Figure 1; Extended Data Table 2). Here, we split the cognitively normal (CN) and MCI cases (with CN A-/N- as the common reference group) and found that the A+/N+ group had higher concentrations of plasma BD-tau, regardless of cognitive status. But in the A+/N- status, no statistically significant elevation of BD-tau was shown, regardless of cognitive status, despite the z-scores being clearly larger than those of the A-/N- (Figure 1). In Biodegmar, we found similar concentrations for serum BD-tau for both A+/N- and A+/N+ cases and argue that BD-tau may have a role as a marker of disease severity (i.e. cognitive status). This is because the A+/N- and A+/N+ groups also had equivalent cognitive performances (Table 1).*

On the question of the BD-tau association with cognition in the A-N- group, we have now included an analysis which shows that serum BD-tau levels are similar between A-/N- CN (SCD cases) and A-/N- CI (MCI + Dementia) in Biodegmar. However, as only 4 cases were CN in the A+ group – no statistical analyses could reliably be carried out to answer this question. Thus, while your comment is of merit, we do not have sufficient cases of CN within the A+ group in the Biodegmar cohort (of where this question is likely better addressed than DDI).

REVIEWER COMMENTS

Reviewer #1 (Remarks to the Author):

I appreciate the detailed updates to the manuscript and responses to the reviewer comments. The manuscript is overall improved and the author responses are thoughtful and in most-to-all cases very reasonable.

1. Unfortunately comment #2 from this reviewer (which is identical to comment #1 from reviewer 2) does not appear to have been well-addressed. The question raised is why BD-tau (proposed as an AD neurodegeneration marker) is elevated in both in N+ and N- individuals in one of the cohorts. Is the conclusion being suggested that BD-tau is only useful in cognitively impaired individuals?
2. For comment #1, the key comparison suggested was to display the profiles of BD-tau in cognitively impaired A+ versus cognitively impaired A- individuals. If BD-tau is also elevated in the latter group, it would argue against its relevance as an AD-specific marker.

Reviewer #2 (Remarks to the Author):

The authors successfully addressed and resolved all the concerns raised in my previous comments.

Reviewer #3 (Remarks to the Author):

As pointed in previous review comments, in one of the two cohorts, BD-tau levels were not different between A+/N+ and A+/N-. This provides the critical discrepancy for the claims in the abstract and even the title to mention the strong findings in this study. For these reviewer's comments, authors replied that Biodegmar cohort did not have enough sample-size to lead the strong conclusion. If so, authors should not mention that BD-tau is neurodegeneration biomarker as a conclusion. (Or, at least, authors should clearly describe the limitation in the manuscript.)

We appreciate the reviewers' comments and apologize for misunderstanding some of the remarks they pointed out in the previous revision. We have updated the manuscript and addressed the main concerns. Here is our reply to their comments.

Reviewer #1 (Remarks to the Author):

I appreciate the detailed updates to the manuscript and responses to the reviewer comments. The manuscript is overall improved, and the author responses are thoughtful and in most-to-all cases very reasonable.

1. Unfortunately comment #2 from this reviewer (which is identical to comment #1 from reviewer 2) does not appear to have been well-addressed. The question raised is why BD-tau (proposed as an AD neurodegeneration marker) is elevated in both in N+ and N- individuals in one of the cohorts. Is the conclusion being suggested that BD-tau is only useful in cognitively impaired individuals?

Reply: Thank you for pointing this out. Unlike cohort 1 (which included mostly cognitively normal and early MCI participants), concentrations of BD-tau in cohort 3 (memory clinic; MCI and dementia participants) were similar in A+/N- and in A+/N+. One plausible explanation for this is that in the Biodegmar memory cohort (cohort 3), the A+/N- and A+/N+ groups shared very similar clinical characteristics. For example, their MMSE and CDR scores were statistically indifferent from each other, meaning participants in the two groups were at the same clinical stages of the disease. Recent reports (<https://pubmed.ncbi.nlm.nih.gov/33439986/>; <https://pubmed.ncbi.nlm.nih.gov/38195725/>) have presented evidence of AD molecular subtypes, of which one is marked with lower CSF t-tau concentrations (A+/N-), blood-brain barrier (BBB) dysfunction and nevertheless high risk of clinical progression, even in comparison with conventional high CSF t-tau AD groups (A+/N+). In more clinically advanced AD (such as in cohort 3), it is possible that blood BD-tau may capture BBB leakage of total-tau to the blood stream, conferring the clinical associations when assessed in blood, regardless of CSF N status. As such, in preclinical and early MCI (cohort 1) BD-tau elevation more closely follow CSF based A/N, whereas at more advanced clinical stages, BD-tau may become increasingly sensitive to disease severity. Indeed, this is particularly emphasized when using blood p-tau/BD-tau as A/N groups, where A+/N+ (regardless of CSF status) robustly associates with clinical impairment in both cohorts. We have now addressed these points in both the result section and discussion.

2. For comment #1, the key comparison suggested was to display the profiles of BD-tau in cognitively impaired A+ versus cognitively impaired A- individuals. If BD-tau is also elevated in the latter group, it would argue against its relevance as an AD-specific marker.

Reply: This is indeed an important point. We have now sourced additional cognitively impaired non-AD cases (normal CSF 42/40 ratio) from cohorts 1 and 3, where we show that BD-tau is significantly higher in cognitively impaired AD vs non-AD cases (Extended Data Figure 4A & B respectively). These results corroborate previously reported findings, where we have demonstrated that blood-based BD-tau is significantly higher in AD versus non-AD.

Reviewer #3 (Remarks to the Author):

As pointed in previous review comments, in one of the two cohorts, BD-tau levels were not different between A+/N+ and A+/N-. This provides the critical discrepancy for the claims in the abstract and even the title to mention the strong findings in this study. For these reviewer's comments, authors replied that Biodegmar cohort did not have enough sample-size to lead the strong conclusion. If so, authors should not mention that BD-tau is neurodegeneration biomarker as a conclusion. (Or, at least, authors should clearly describe the limitation in the manuscript.)

Reply: Thank you. Please see reply to comment 1 & 2 of reviewer 1, where we believe we now have addressed these points at the best of our abilities, including AD/non-AD BD-tau differences. We have now updated the manuscript with the corresponding text and added to the limitations the lack of CN A+ participants in cohort 3.

Finally, we want to thank the reviewers for their comments which have greatly help us to improve the quality of the manuscript. We will be happy to address any further comments or questions.

REVIEWERS' COMMENTS

Reviewer #1 (Remarks to the Author):

I appreciate the author's responses to the remaining comments. The work remains of high quality overall. I had a very difficult time making sense out of the updated response to comment #1 (the same issue raised by reviewer 3), but it is certainly possible that cohort differences in characteristics may account for the different results. However, I agree with reviewer 3's summation of the issue - it would be very challenging for readers to see a title proposing BDtau as a neurodegeneration biomarker only for the data in one of the cohorts (a memory clinic cohort at that) arguing against it. I would suggest the title and conclusions be modified or at least softened/nuanced in language (with much more addressing of this issue in the Discussion than the very few remarks of limitations as current) to better represent the findings.

Reviewer #3 (Remarks to the Author):

Thank you for the update of manuscript. The authors addressed the concerns raised in my comments.

We appreciate the comments and suggestions received. Please find our comments to the latest remarks below.

Reviewer #1 (Remarks to the Author):

I appreciate the author's responses to the remaining comments. The work remains of high quality overall. I had a very difficult time making sense out of the updated response to comment #1 (the same issue raised by reviewer 3), but it is certainly possible that cohort differences in characteristics may account for the different results. However, I agree with reviewer 3's summation of the issue - it would be very challenging for readers to see a title proposing BDtau as a neurodegeneration biomarker only for the data in one of the cohorts (a memory clinic cohort at that) arguing against it. I would suggest the title and conclusions be modified or at least softened/nuanced in language (with much more addressing of this issue in the Discussion than the very few remarks of limitations as current) to better represent the findings.

Reply: Thank you so much for your comments. We have edited the text and extended the discussion to address the main concerns regarding the findings in Biodegmar (cohort 3).

Reviewer #3 (Remarks to the Author):

Thank you for the update of manuscript. The authors addressed the concerns raised in my comments.

Reply: Thank you!

Finally, we would like to thank the editors and reviewers because thanks to their feedback and suggestions we have significantly improved the quality of our manuscript.